# Selection of multi-model ensemble of GCMs for the simulation of precipitation, maximum and minimum temperature based on spatial assessment metrics

Kamal Ahmed[1, 2], Dhanapala A. Sachindra[3], Shamsuddin Shahid[1], Mehmet C. Demirel[4], Eun-Sung Chung[5]

[1]Faculty of Civil Engineering, Universiti Teknologi Malaysia (UTM), Johor Bahru, 81310, Malaysia
[2]Faculty of Water Resource Management, Lasbela University of Agriculture, Water and Marine Sciences, Balochistan, 90150, Pakistan
[3]Institute for Sustainability and Innovation, College of Engineering and Science, Victoria University, P.O. Box 14428, Melbourne, Victoria 8001, Australia
[4]Department of Civil Engineering, Istanbul Technical University, 34469 Maslak, Istanbul, Turkey
[5]Department of Civil Engineering, Seoul National University of Science and Technology, Seoul, 01811, Republic of Korea

*Correspondence to*: Eun-Sung Chung (eschung@seoultech.ac.kr)

**Abstract.** The climate modelling community has trialled a large number of metrics for evaluating temporal performance of General Circulation Models (GCMs), while very little attention has been given to the assessment of their spatial performance which is equally important. This study evaluated the performance of 36 Coupled Model Intercomparison Project 5 (CMIP5) GCMs in relation to their skills in simulating mean annual, monsoon, winter, pre-monsoon, and post-monsoon precipitation, maximum and minimum temperature over Pakistan using state-of-the-art spatial metrics; SPAtial EFficiency, Fractions Skill Score, Goodman–Kruskal's lambda, Cramer's V, Mapcurves, and Kling-Gupta efficiency for the period 1961-2005. The multi-model ensemble (MME) precipitation, maximum and minimum temperature data were generated through the intelligent merging of simulated precipitation, maximum and minimum temperature of selected GCMs employing Random Forest (RF) regression and Simple Mean (SM). The results indicated some differences in the ranks of GCMs for different spatial metrics. The overall ranks indicated NorESM1-M, MIROC5, BCC-CSM1-1 and ACCESS1-3 as the best GCMs in simulating the spatial patterns of mean annual, monsoon, winter, pre-monsoon, and post-monsoon precipitation, maximum and minimum temperature over Pakistan. MME precipitation, maximum and minimum temperature generated based on the best performing GCMs showed more similarities with observed precipitation, maximum and minimum temperature compared to precipitation, maximum and minimum temperature simulated by individual GCMs. The MMEs developed using RF displayed better performance than the MMEs-based on SM. Multiple spatial metrics have been used for the first time for selecting GCMs based on their capability to mimic the spatial patterns of annual and seasonal precipitation, maximum and minimum temperature. The approach proposed in the present study can be extended to any number of GCMs and climate variables and applicable to any region for the suitable selection of an ensemble of GCMs to reduce uncertainties in climate projections.

## 1 Introduction

Climate change is a complex, multidimensional phenomenon that is being critically studied over the last few decades (Byg and Salick, 2009;Cameron, 2011). The changes in climate are mostly observed by studying the variations in precipitation and temperature regimes (Sheffield and Wood, 2008). Several studies reported increase in severity and frequency of droughts (Ahmed et al., 2019a), floods (Wu et al., 2014), heatwaves (Perkins-Kirkpatrick and Gibson, 2017) and decrease in severity and frequency of cold snaps (Wang et al., 2016) in the recent years which are indicative of abrupt variations in the precipitation and temperature regimes. According to the Intergovernmental Panel on Climate Change (IPCC) 5[th] Assessment Report (AR5), the average global land and ocean surface air temperature has risen by around 0.72℃ (0.49–0.89℃) during 1951–2012. It is projected that it will further increase by 1.8 ℃ to 4 ℃ by the end of the 21[st] century (IPCC, 2014). The climate modelling community has widely agreed that the sharp temperature rise in the post-industrial revolution era is significantly affecting the global hydrologic cycle (Sohoulande Djebou and Singh, 2015;Evans, 1996). The spatiotemporal variations in the global hydrologic cycle are influential on the humans and the environment. Therefore, it is important to study the variations in spatiotemporal patterns of climate variables such as precipitation and temperature (Akhter et al., 2017).

General Circulation Models (GCMs) are principally utilised to simulate and project climate on a global scale (Pour et al., 2018;Sachindra et al., 2014). Over the years, a large number of GCMs have been developed and used for the simulation and projection of the global climate. The Coupled Model Intercomparison Project Phase 5 (CMIP5) is a set of GCMs available from the IPCC AR5 (Taylor et al., 2012). The CMIP5 GCMs showed significant improvements in climate simulations compared to its previous generation of CMIP3 models (Gao et al., 2015;Kusunoki and Arakawa, 2015). Currently, over 50 GCMs are available in the CMIP5 suite with different spatial resolutions (Hayhoe et al., 2017). Human and computational resources pose a restriction on the size of the sub-set of GCMs used in a climate change impact assessment (Herger et al., 2018). Sa'adi et al. (2017), Salman et al. (2018a), Pour et al. (2018) and Khan et al. (2018a) reported that a multi-model ensemble (a sub-set) of GCMs selected considering their skills in reproducing past observed characteristics of climate can reduce the GCM associated uncertainties in climate change impact assessment. The multi-model ensembles (MME) also enhance the reliability of projection using information from several sources or GCMs (Pavan and Doblas-Reyes, 2000;Knutti et al., 2010).

The methods used for the generation of MMEs are broadly divided into two groups; (1) simple composite method (SCM) and (2) weighted ensemble method (WEM) (Wang et al., 2018). In SCM all ensemble members are equally weighted while in the WEM, ensemble members are weighted according to their performance in simulating the past climate (Wang et al., 2018;Oh and Suh, 2017;Giorgi and Mearns, 2002). The SCM is relatively simple to apply and found to perform better than individual GCMs (Weigel et al., 2010;Acharya et al., 2013;Wang et al., 2018). However, WEM is preferred as it can remove the systematic biases and improve the prediction capability since higher weights are assigned to better GCMs (Krishnamurti et al., 1999;Krishnamurti et al., 2000). Salman et al. (2018a) reported that the prediction capability of an MME improves if it

is based on the WEM method. Thober and Samaniego (2014) also showed that sub-ensembles generated using WEM has a better capability to capture the historical characteristics of precipitation and temperature extremes. The performances of MMEs depend on the performance of ensemble members in simulating historical climate (Pour et al., 2018). Therefore, selection of a sub-ensemble is a major challenge in climate change modelling.

Numerous endeavours have been made to examine the adequacy of climate models in simulating various climate variables (e.g. precipitation) (McMahon et al., 2015;Gu et al., 2015). Smith et al. (1998) stated that selection of an appropriate set of GCMs in a climate change impact assessment can be achieved considering 4 criteria; (1) Vintage - only the latest generation GCMs are considered, (2) Spatial resolution - fine resolution GCMs are preferred over coarser ones, (3) Validity - performances of GCMs are considered, and (4) Representativeness - an ensemble of GCMs covering a wide range of

projections of a climate variable (e.g. precipitation) is considered. In the above criteria, assessment and selection of GCMs based on their validity is the most widely adopted criterion where GCMs are ranked and selected according to their skill in simulating observed past climate (Mendlik and Gobiet, 2016).

A wide variety of methods has been used to assess climate models based on their ability to simulate the observed historical climate (past performance) such as reliability ensemble averaging approach (Giorgi and Mearns, 2002), relative entropy

(Shukla et al., 2006), Bayesian approach (Min and Hense, 2006;Tebaldi et al., 2005;Chandler, 2013), probability density function (Perkins et al., 2007), hierarchical ANOVA models (Sansom et al., 2013), clustering (Knutti et al., 2013), correlation (Xuan et al., 2017;Jiang et al., 2015), and symmetrical uncertainty (Salman et al., 2018a). Johnson and Sharma (2009) assessed the performance of GCMs in replicating inter-annual variability. Thober and Samaniego (2014) evaluated the performance of GCMs in reproducing extreme indices of precipitation and temperature. Apart from that, some studies

combined several performance measures such as root means square error, mean absolute error, correlation coefficient, and skill scores into one performance index to assess the accuracy of GCMs in reproducing past climate (Gu et al., 2015;Barfus and Bernhofer, 2015;Gleckler et al., 2008;Wu et al., 2016;Ahmadalipour et al., 2017;Raju et al., 2017). Moreover, the past performance assessment of GCM is performed at different temporal scales; daily (Perkins et al., 2007), monthly (Raju et al., 2017), seasonal (Ahmadalipour et al., 2017) and annual (Murphy et al., 2004). Besides temporal scales, a number of studies

ranked GCMs based on spatial areal average (Ahmadalipour et al., 2017;Abbasian et al., 2019), while some studies considered GCM performance at all the grid points covering the study area (Raju et al., 2017;Salman et al., 2018a).

It is also observed in the literature that there is no consensus on the choice of the GCM selection approach and temporal scale at which the performance assessment is done. Raäisaänen (2007), Smith and Chandler (2010) and McMahon et al. (2015) also argued that there is no universally accepted criterion for the assessment of GCMs. However, McMahon et al.

(2015) reported that GCM simulations at the annual time scale can better reproduce long-term mean statistics compared to that at daily time scale. Gleckler et al. (2008) stated that assessment of GCMs with respect to a climate variable like precipitation over multiple time scales or seasons may provide vital information to water resources managers especially in the regions where climate variability is high. Moreover, Raju et al. (2017) and Salman et al. (2018a) demonstrated that GCM assessment provides more useful information when the evaluation is conducted at individual grid points covering the study

area of interest. Selection of GCMs based on their performance at individual grid points over a region does not guarantee their capability to simulate spatial patterns of regional climate. It is expected that GCMs should be able to capture the spatial pattern of major features of the climate of a region such as a monsoon and western disturbances. Koch et al. (2018) and Demirel et al. (2018) argued that climate modelling community is mostly focused on the temporal performance of GCMs

and ignores explicit assessment of their spatial performance which is also equally important. They also emphasized on the importance of the use of multiple spatial metrics for GCM performance assessment. Furthermore, the metrics should be insensitive to the units of the variables compared.

Overall, review of literature revealed that several studies (Khan et al., 2018a;Pour et al., 2018;Salman et al., 2018a;Raju et al., 2017) assessed the performance of GCMs considering several grid points over the whole study area; however they

ignored the capability of GCMs to replicate the spatial patterns. Spatial patterns of GCMs provide a better understanding of the occurrences of hydro-climatic phenomena such as precipitation distributions, floods and droughts. Therefore, it is imperative to assess the skills of GCMs in replicating the historical spatial patterns of climate variables. Within this framework, the current study hypothesized that the sub-ensemble members identified based on their ability to mimic the spatial pattern of observed precipitation, maximum and minimum temperature of a region can be used for the generation of a

reliable MME for precipitation, maximum and minimum temperature for that region. This study for the first time, employed six state-of-the-art spatial performance metrics; SPAtial EFficiency metric (SPAEF) (Demirel et al., 2018), Fractions Skill Score (FSS) (Roberts and Lean, 2008), Goodman–Kruskal's lambda (Goodman and Kruskal, 1954), Cramer's V (Cramér, 1999), Mapcurves (Hargrove et al., 2006), and Kling-Gupta efficiency (KGE) (Gupta et al., 2009) for the assessment of performance of 36 CMIP5 GCM in simulating observed annual (Jan to Dec), monsoon (Jun to Sep), winter (Dec to Mar),

pre-monsoon (Apr to May), and post-monsoon (Oct to Nov) precipitation, maximum and minimum temperature over Pakistan. These metrics were selected based on their recent applications in several spatial performance assessment studies (Demirel et al., 2018;Koch et al., 2018;Rees, 2008). Then based on the above spatial performance metrics the most skilful GCMs were identified and hence multi-model ensemble (MME) means of precipitation, maximum and minimum temperature using Simple Mean (SM) and Random Forest (RF) were generated.

**2 Study Area and Datasets**

**2.1 Study area**

As shown in Figure 1, Pakistan located in south Asia shares its border with India in the east, China in the north, Afghanistan and Iran in the west and the Arabian Sea in the south. Pakistan has a rugged topography ranging from 0 m in the south to 8572 m in the north. Figure 2 which is based on the study by Ahmed et al. (2019d) shows that a large area of Pakistan

experiences an arid climate, followed by semi-arid climate, while a small area in the southwest region experiences hyper-arid climate. However, a small area in the northernmost region of the country experiences sub-humid to humid climate.

Pakistan receives summer monsoon precipitation during the period June-September and winter precipitation during the period December-March. Besides that, there are two intermediate rainy seasons called the pre-monsoon and the post-monsoon during the periods April-May and October-November, respectively (Sheikh, 2001). The bulk of the summer precipitation is caused by the monsoon winds that arise from the Bay of Bengal while westerly disturbances in the Mediterranean Sea are responsible for the winter precipitation. The average precipitation in Pakistan widely varies from southwest to northern parts in the range of < 100 to > 1000 mm/year (based on data from 1961 to 2010). Since the country is mostly characterized by arid and semi-arid climate; the bulk of the country receives precipitation less than 500 mm/year while only a very limited area in the north receives more than 1,000 mm/year of precipitation (Ahmed et al., 2017). The average temperature of the country varies from 0° C in the northern region to 32° C in the southern region (Khan et al., 2018b).

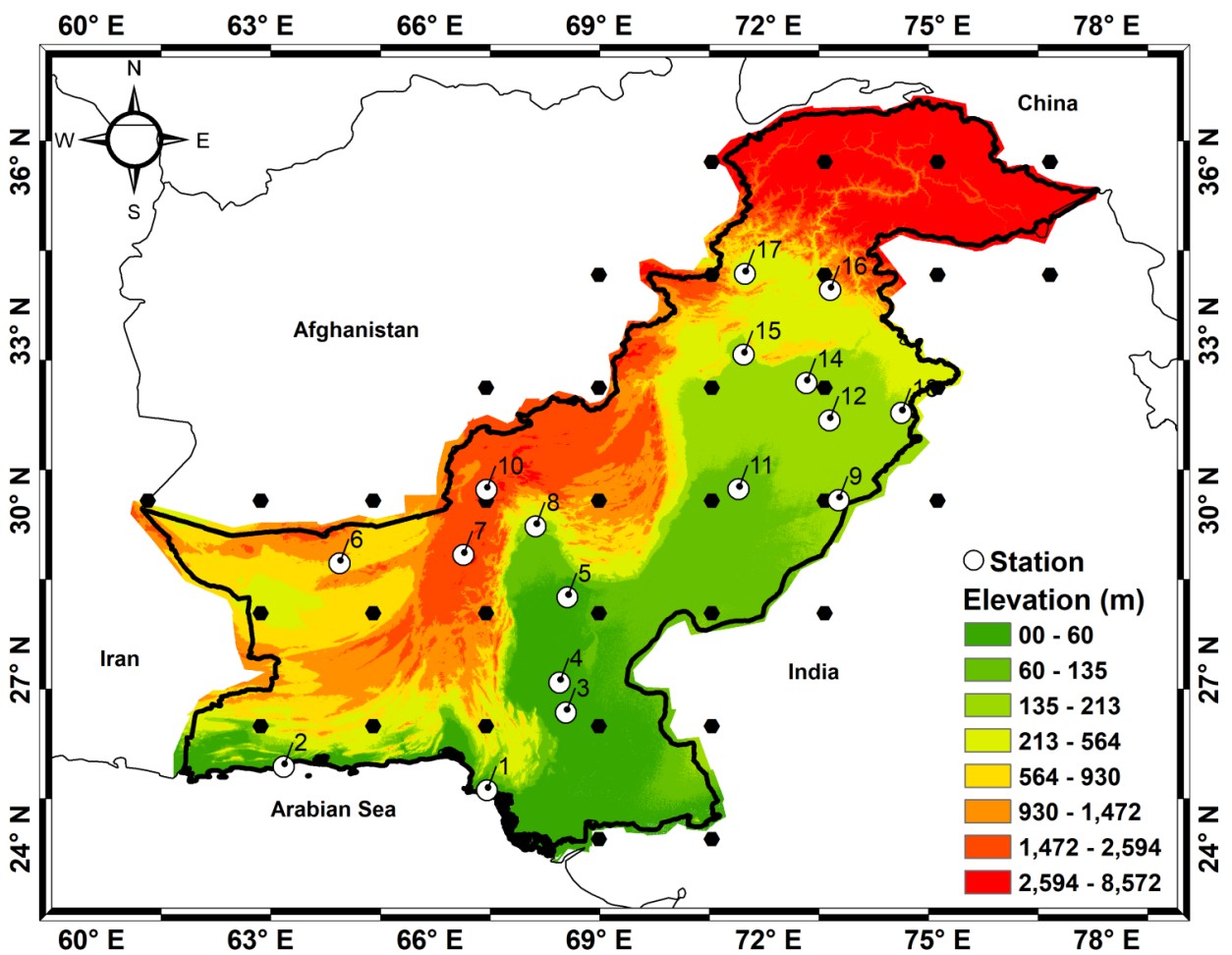

**Figure 1.** The location of Pakistan in central-south Asia and the GCM grid points over the country along with the locations of precipitation and temperature observation stations. The names of the stations are given in Table 2.

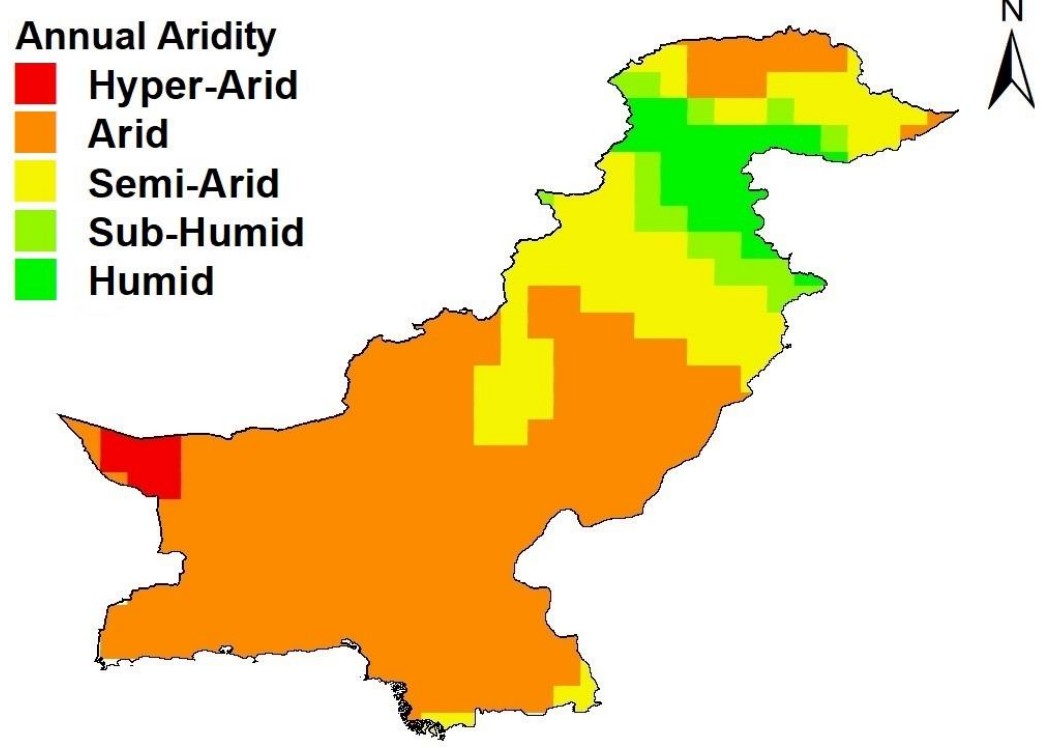

**Figure 2.** Aridity classification of Pakistan (adopted from Ahmed et al. (2019d))

**2.2 Datasets**

**2.2.1 Gridded Precipitation and Temperature data**

The lack of long records of climate observations with extensive spatial coverage is a major issue in hydro-climatological investigations in many regions. As a solution to this problem, gridded datasets based on observations and various interpolation and data assimilation techniques have been created (Kishore et al., 2015). In this investigation, gridded monthly precipitation data of the Global Precipitation Climatology Center (GPCC) (Schneider et al., 2013) (dwd.de/EN/ourservices/gpcc/gpcc.html), and gridded monthly maximum and minimum temperature data of Climatic Research Unit (CRU) of East Anglia University (https://crudata.uea.ac.uk/cru/data/hrg/) (Harris et al., 2014) were used as the surrogates of observed precipitation, maximum and minimum temperature respectively for the period 1961-2005. GPCC precipitation and CRU temperature data are available at a spatial resolution of 0.5°. As stated in the existing literature GPCC and CRU data are of high quality (Shiru et al., 2018;Salman et al., 2018b) and have an excellent seamless spatial and temporal coverage (Spinoni et al., 2014). Most importantly, GPCC precipitation and CRU temperature data have shown

correlations above 0.80 with observed precipitation, maximum and minimum temperature over Pakistan (Ahmed et al., 2019c).

**2.2.2 GCM precipitation and Temperature data**

Monthly precipitation data simulated by the 36 CMIP5 GCMs for ensemble run r1i1p1 were extracted from the IPCC data distribution center (http://www.ipcc-data.org/sim/gcm_monthly/AR5/Reference-Archive.html) for the period 1961-2005. The modelling centres, names of GCMs and spatial resolution of each of the selected GCMs are provided in Table 1. In order to have a common spatial resolution, precipitation ($P$), maximum temperature ($T_{max}$) and minimum temperature ($T_{min}$) data obtained from different GCMs and GPCC and CRU databases were interpolated into a common 2°×2° grid using bilinear interpolation.

**Table 1.** CMIP5 GCMs considered in this study.

| Country | Modelling Centre | Model Name | Resolution in arc degrees (Lat) | Resolution in arc degrees (Lon) |
|---|---|---|---|---|
| Australia | Commonwealth Scientific and Industrial Research Organization/Bureau of Meteorology | ACCESS1-0 | 1.25 | 1.875 |
| | | ACCESS1-3 | 1.25 | 1.875 |
| | Commonwealth Scientific and Industrial Research Organization/Queensland Climate Change Centre of Excellence | CSIRO-Mk3-6-0 | 1.8653 | 1.875 |
| Canada | Canadian Centre for Climate Modelling and Analysis | CanESM2 | 2.7906 | 2.8125 |
| China | Beijing Climate Center | BCC-CSM1.1(m) | 2.7906 | 2.8125 |
| | | BCC-CSM1-1 | 2.7906 | 2.8125 |
| | Beijing Normal University | BNU-ESM | 2.7906 | 2.8125 |
| | Institute of Atmospheric Physics, Chinese Academy of Sciences | FGOALS-g2 | 2.7906 | 2.8125 |
| | The First Institute of Oceanography, SOA | FIO-ESM | 2.81 | 2.78 |
| France | Institut Pierre-Simon Laplace | IPSL-CM5A-LR | 1.8947 | 3.75 |
| | | IPSL-CM5A-MR | 1.2676 | 2.5 |
| | | IPSL-CM5B-LR | 1.8947 | 3.75 |
| | Centre National de Recherches Météorologiques, Centre Européen de Recherche et de Formation Avancée en Calcul Scientifique | CNRM-CM5 | 1.4008 | 1.40625 |

| Country | Institution | Model | Value 1 | Value 2 |
|---|---|---|---|---|
| Germany | Max Planck Institute for Meteorology | MPI-ESM-LR | 1.8653 | 1.875 |
| | | MPI-ESM-MR | 1.8653 | 1.875 |
| Italy | Centro Euro-Mediterraneo sui Cambiamenti Climatici | CMCC-CM | 0.7484 | 0.75 |
| | | CMCC-CMS | 3.7111 | 3.75 |
| Japan | Atmosphere and Ocean Research Institute (The University of Tokyo), National Institute for Environmental Studies, and Japan Agency for Marine-Earth Science and Technology | MIROC5 | 1.4008 | 1.40625 |
| | | MIROC-ESM | 2.7906 | 2.8125 |
| | | MIROC-ESM-CHEM | 2.7906 | 2.8125 |
| | Meteorological Research Institute | MRI-CGCM3 | 1.12148 | 1.125 |
| Netherlands/Ireland | EC-EARTH consortium published at Irish Centre for High-End Computing | EC-EARTH | 1.1215 | 1.125 |
| Norway | Bjerknes Centre for Climate Research, Norwegian Meteorological Institute | NorESM1-M | 1.8947 | 2.5 |
| Russia | Russian Academy of Sciences, Institute of Numerical Mathematics | inmcm4 | 1.5 | 2 |
| South Korea | National Institute of Meteorological Research, Korea Meteorological Administration | HadGEM2-AO | 1.25 | 1.875 |
| UK | Met Office Hadley Centre | HadGEM2-CC | 1.25 | 1.875 |
| | | HadGEM2-ES | 1.25 | 1.875 |
| USA | National Center for Atmospheric Research | CCSM4 | 0.9424 | 1.25 |
| | | CESM1-BGC | 0.9424 | 1.25 |
| | | CESM1-CAM5 | 0.9424 | 1.25 |
| | | CESM1-WACCM | 1.8848 | 2.5 |
| | Geophysical Fluid Dynamics Laboratory | GFDL-CM3 | 2 | 2.5 |
| | | GFDL-ESM2G | 2.0225 | 2 |
| | | GFDL-ESM2M | 2.0225 | 2.5 |
| | NASA/GISS (Goddard Institute for Space Studies) | GISS-E2-H | 2 | 2.5 |
| | | GISS-E2-R | 2 | 2.5 |

## 3 Methodology

In this study, GCMs for annual, monsoon, winter, pre-monsoon and post-monsoon $P$, $T_{max}$ and $T_{min}$ were first ranked separately (individual ranking) using six spatial performance measures; SPAEF, FSS, Lambda, Cramer-V, Mapcurves, and KGE. Then a comprehensive rating metric (RM) (Jiang et al., 2015) was used to rank the GCMs considering the individual ranks determined corresponding to all above spatial performance measures. The RM values of GCMs obtained for each variable were combined for deriving the overall ranks of GCMs. Finally, a sub-set of GCMs (MME) based on the overall ranks was selected and $P$, $T_{max}$ and $T_{min}$ data for the MME were derived. The procedure used for the ranking, identification of the ensemble of GCMs and derivation of $P$, $T_{max}$ and $T_{min}$ data from the multi-model ensemble of GCMs is outlined as follows.

1. All GCM simulated past $P$, $T_{max}$ and $T_{min}$ data for the period 1961-2005 were remapped to a common grid with a 2°×2° resolution.

2. SPAEF, FSS, Lambda, Cramer-V, Mapcurves, and KGE were individually applied to annual, monsoon, winter, pre-monsoon and post-monsoon $P$, $T_{max}$ and $T_{min}$ data for the period 1961-2005.

3. The goodness of fit (GOF) estimated by SPAEF, FSS, Lambda, Cramer-V, Mapcurves, and KGE for annual, monsoon, winter, pre-monsoon and post-monsoon $P$, $T_{max}$ and $T_{min}$ were used to rank the GCMs separately.

4. Comprehensive rating metrics (RM) was used to combine the ranks of GCMs determined by the above spatial performance measures separately for $P$, $T_{max}$ and $T_{min}$.

5. RM was again used to derive the overall ranks of GCMs considering $P$, $T_{max}$ and $T_{min}$ together for the entire study area.

6. The four-top ranked GCMs based on their overall ranks in replicating annual, monsoon, winter, pre-monsoon and post-monsoon $P$, $T_{max}$ and $T_{min}$ were identified.

7. Simple Average (SM) and Random Forest (RF) were used to generate MME $P$, $T_{max}$ and $T_{min}$ means with the $P$, $T_{max}$ and $T_{min}$ simulated by the four-top ranked GCMs identified in step 6.

8. Finally, the spatial patterns of MME $P$, $T_{max}$ and $T_{min}$ generated using SM and RF were validated by visually comparing them with the spatial patterns of observed $P$, $T_{max}$ and $T_{min}$.

Details of the methods and the determination of the best performing ensemble of GCMs are provided in the following sections.

**3.1 Accuracy Assessment of Gridded Precipitation and Temperature Data**

The accuracy of gridded GPCC precipitation data and CRU temperature data was assessed by comparing them with the observed station data using normalised root mean square error (NRMSE) and modified index of agreement (*md*). NRMSE is a non-dimensional form of root mean square error (RMSE) which is derived by normalizing RMSE by the range of observations. NRMSE is more reliable than RMSE in comparing model performance when the model outputs are in different units or the same unit but with different orders of magnitude (Willmott, 1982). NRMSE can have any positive value, however, values closer to 0 are preferred as they denote smaller errors (Chen and Liu, 2012). In this study, NRMSE was calculated Eq. 1.

$$NRMSE = \frac{\left[\frac{1}{N}\sum_{i=1}^{N}\left(x_{sim,i} - x_{obs,i}\right)^2\right]^{1/2}}{x_{max} - x_{min}} \tag{1}$$

Where $x_{sim,i}$ and $x_{obs,i}$ refer to the $i^{th}$ value in the gridded and observed time series of the climate variable (i.e. precipitation or temperature) respectively, and $N$ is the number of data points in each time series.

The '*md*' shown in Eq. 2 is widely used to estimate the agreement between observed and gridded data of climate variables (Noor et al., 2019;Ahmed et al., 2019b). It varies between 0 (no agreement) and 1 (perfect agreement) (Willmott, 1981).

$$md = 1 - \frac{\sum_{i=1}^{n}\left(x_{obs,i} - x_{sim,i}\right)^j}{\sum_{i=1}^{n}\left(\left|x_{sim,i} - \overline{x_{obs}}\right| + \left|x_{obs,i} - \overline{x_{obs}}\right|\right)^j} \tag{2}$$

Where $x_{sim,i}$ and $x_{obs,i}$ are the $i^{th}$ value in the gridded data and observed data series of a climate variable.

**3.2 GCM Performance Assessment**

SPAEF, FSS, Lambda, Cramer-V, Mapcurves, and KGE were individually applied on each year from 1961 to 2005 of mean annual, monsoon, winter, pre-monsoon, and post-monsoon $P$, $T_{max}$ and $T_{min}$. Later, the GOF values of each year were temporally averaged to obtain a value for the entire study area. The details of the metrics are given below.

### 3.2.1 SPAtial EFficiency metric

SPAtial EFficiency metric (SPAEF), proposed by Demirel et al. (2018) is a robust spatial performance metric which considers three statistical measures (1) Pearson correlation, (2) coefficient of variation and (3) histogram overlap, in the assessment of GOF of a model. The major advantage of SPAEF is that it combines the information derived from the above three independent statistical measures into one metric. The SPAEF values between past observed GPCC $P$, CRU $T_{max}$ and $T_{min}$ and GCM simulated $P$, $T_{max}$ and $T_{min}$ were calculated using Eq. 3. In Eq. 3, α is the Pearson correlation coefficient between observed and GCM simulated data, β is the spatial variability and γ is the overlap between the histograms of observed and GCM simulated data.

$$SPAEF = 1 - \sqrt{(\alpha - 1)^2 + (\beta - 1)^2 + (\gamma - 1)^2} \qquad (3)$$

Equations 4 and 5 show the procedure for β and γ calculations respectively (for Pearson correlation (α) refer to (Pearson, 1948). In Eq. 4 $\sigma_G$ and $\sigma_O$ refer to standard deviation of GCM simulated and observed data respectively and $\mu_G$ and $\mu_O$ refer to mean of GCM simulated and observed data respectively.

$$\beta = \frac{\left(\frac{\sigma_G}{\mu_G}\right)}{\left(\frac{\sigma_O}{\mu_O}\right)} \qquad (4)$$

In Eq. 5, $K$, $L$ and $n$ refer to histograms value of observations, histograms value of GCM simulations and the number of bins in a histogram.

$$\gamma = \frac{\sum_{j=1}^{n} \min(K_j, L_j)}{\sum_{j=1}^{n} K_j} \qquad (5)$$

The SPAEF can have a value between $-\infty$ and 1, where a value closer to 1 indicates higher spatial similarity between the observations and model simulations (Koch et al., 2018). A code written in MATLAB environment was used for calculating SPAEF values (Demirel et al., 2018).

### 3.2.2 Fractions Skill Score

The Fractions Skill Score (FSS) proposed by (Roberts and Lean, 2008) is another measure used for the assessment of spatial agreement between model simulations and observations. *FSS* varies between 0 and 1 where a value closer to 1 refers to a higher agreement between observed and simulated data. In this study, *FSS* between observed and GCM simulated data was computed using Eq. 6.

$$FSS = 1 - \frac{MSE_{(n)}}{MSE_{(n)ref}}$$
(6)

In Eq. 6 MSE refers mean square error and is calculated using Eq. 7 and 8.

$$MSE_{(n)} = \frac{1}{N_x N_y} \sum_{i=1}^{N_x} . \sum_{j=1}^{N_y} \left[ O_{(n)i,j} - M_{(n)i,j} \right]^2$$
(7)

$$MSE_{(n)ref} = \frac{1}{N_x N_y} \left[ \sum_{i=1}^{N_x} \sum_{j=1}^{N_y} O_{(n)i,j}^2 + \sum_{i=1}^{N_x} \sum_{j=1}^{N_y} M_{(n)i,j}^2 \right]$$
(8)

In Eq. 7 and 8 $N_x$ and $N_y$ are the number of columns and rows in an observed or simulated map of a climate variable respectively, $O$ and $M$ are observed and simulated data fractions respectively. The "verification" package (Pocernich, 2006) written in R programming language was employed in this study for estimating FSS values.

### 3.2.3 Goodman–Kruskal's lambda

Goodman–Kruskal's lambda also known as Lambda coefficient ($\lambda$) is used to measure the nominal/categorical association between categorical maps (Goodman and Kruskal, 1954). Lambda coefficient ($\lambda$) varies between 0 and 1, where a value closer to 1 refers to a higher similarity between the map of model simulations and that of observations of *P*, *T<sub>max</sub>* and *T<sub>min</sub>*. The Lambda ($\lambda$) coefficient was calculated using Eq. 9, where $max_j$ is the number of classes (categories) in the observed and simulated maps, $c_{ij}$ is a contingency matrix (describes the relationships between the data classes), $i$ and $j$ are the classes in observed and simulated maps, $m$ represents the number of classes in the observed and simulated maps respectively. In the present study, seven classes in the contingency matrix were used by following the study by Demirel et al. (2018). The "DescTools" package (Signorell, 2016) written in R programming language was employed in this study for estimating the nominal/categorical association between observed and simulated maps.

$$\lambda = \frac{\sum_{i=1}^{m} max_j c_{ij} - max_j \sum_{i=1}^{m} c_{ij}}{N - max_j \sum_{i=1}^{m} c_{ij}}$$
(9)

### 3.2.4 Cramer's V

Cramer's V (Cramér, 1999) statistic is a Chi-square-test-based measure which is used in assessing spatial agreement between observations and model simulations (Zawadzka et al., 2015). Its value ranges between 0 and 1 and value closer to 1 refers to a better agreement between the simulated and observed maps of the climate variable. Cramer's V was calculated using Eq. 10.

$$V = \sqrt{\frac{x^2}{N(\min(m,n)-1)}} \tag{10}$$

where, $x^2$ is Chi-Square, $N$ is the grand total of observations, $m$ is the number of rows and $n$ is the number of columns. In this exercise $m = 42$ (number of rows of data) and $n = 2$ (observed and modelled precipitation). The "DescTools" package (Signorell, 2016) written in R programming language was employed in this study for calculating Cramer's V values.

### 3.2.5 Mapcurves

Mapcurves is another statistical measure, developed by Hargrove et al. (2006) for the measurement of similarity between categorical maps. Mapcurves quantifies the degree of concordance between two maps. The value of Mapcurves can vary from 0 to 1 (perfect agreement). In the present study, the degree of concordance between the historical observed $P$, $T_{max}$ and $T_{min}$ map and each of the GCM simulated $P$, $T_{max}$ and $T_{min}$ maps was determined using Eq. 11 where, $MC_X$ refers the Mapcurves value, A is the total area of a given class X on the map being compared, B is the total area of a given class Y on the observed map, C is the area of intersection between X and Y when the maps are overlaid and $n$ is the number of classes in the observed map.

$$MC_X = \sum_{Y=1}^{n} \left[ \left( \frac{C}{A} \cdot \frac{C}{B} \right) \right] \tag{11}$$

In this study, the function "*mapcurves(x,y)*" available in "sabre" package (Nowosad and Stepinski, 2018) written in R programing language was used for estimating mapcurves values. In that equation $x$ and $y$ are vectors represent the categorical values of historical observed data (e.g. GPCC precipitation) and categorical values of simulated data by a GCM, respectively.

### 3.2.6 Kling-Gupta efficiency

Kling-Gupta efficiency (KGE) is a GOF test developed by Gupta et al. (2009), for the model performance assessment. KGE considers three statistical measures (1) Pearson correlation, (2) variability ratio and (3) bias ratio, in the assessment of model performance. In the present study, KGE was calculated between historical observed data and GCM simulated data using Eq. 12. KGE values can range between –infinity and 1, where values close to 1 are preferred.

$$KGE = 1 - \sqrt{(\alpha_P - 1)^2 + (\beta_P - 1)^2 + (\gamma_{RP} - 1)^2} \tag{12}$$

In Eq. 12, $\alpha_P$ is the Pearson correlation (Pearson, 1948) between observed and GCM simulated data, $\beta_P$ is the bias ratio, and $\gamma_{RP}$ is the variability ratio. Equations 13 and 14, show the calculation of $\beta_P$ and $\gamma_{RP}$ respectively.

$$\beta_P = \frac{\mu_G}{\mu_O} \tag{13}$$

In Eq. 13, $\mu_G$ and $\mu_O$ refer to mean of GCM simulated and observed data respectively.

$$\gamma_{RP} = \frac{CV_G}{CV_O} = \frac{\left(\frac{\sigma_G}{\mu_G}\right)}{\left(\frac{\sigma_O}{\mu_O}\right)} \tag{14}$$

In Eq. 14, $CV_G$ and $CV_O$ refer to coefficient of variation of GCM simulated and observed data respectively.

### 3.3 Comprehensive Rating Metrics

The ranking of GCMs with respect to a given climate variable using one single GOF measure is a relatively simple task. However, the ranking of GCMs becomes more challenging when multiple GOF measures are used with multiple climate variables, as different GCMs may display different degrees of accuracies for different GOF measures and climate variables. In such a case, an information aggregation approach that combines information from several GOF measures can be used. In this study, a comprehensive rating metric (Chen et al., 2011) was used to obtain the overall ranks of GCMs. The overall

ranks of GCMs based on different GOFs were obtained for each season separately using Eq. 15.

$$RM = 1 - \frac{1}{nm} \sum_{i=1}^{n} rank_i \tag{15}$$

In Eq. (15), *n* refers to the number of GCMs, *m* refers to the number of metrics or seasons and *i* refers to the rank of a GCM

based on *i*th GOF. A value of RM near to 1 refers to a better GCM in terms of its ability to mimic the spatial or temporal characteristics of observations.

### 3.4 Identification of Ensemble Members

The uncertainties in climate projections which arise from GCM structure, assumptions and approximations, initial conditions, and parameterization can be reduced by identifying an ensemble of better performing GCMs (Kim et al., 2015). Lutz et al.

(2016) reported that one or a small ensemble of GCMs is suitable for climate change impact assessment. A number of studies (Weigel et al., 2010;Miao et al., 2012) have suggested that one GCM is not enough to assess the uncertainties

associated with the future climate. Therefore, identification of an ensemble of GCMs is a necessity in climate change impact assessments. In the present study, four top-ranked GCMs were considered for the development of MMEs for $P$, $T_{max}$ and $T_{min}$. The review of the literature revealed that there is no well-defined guideline on the selection of the optimum number of GCMs for the MME and most of the studies considered the first three to ten GCMs ranked according to the descending order of their performance for the MME. For instance, in the study by Xuan et al. (2017) over Zhejiang, China, ten top-ranked GCMs for an MME for precipitation were used. In another study over China, Jiang et al. (2015) developed MMEs for daily temperature extremes using the five top-ranked GCMs. In a study over Pakistan, Khan et al. (2018a) considered six common GCMs that appeared in the lists of ten top-ranked GCMs for daily temperature and precipitation. Ahmadalipour et al. (2017) used the four top-ranked GCMs for simulating daily precipitation and temperature over the Columbia River Basin in the Pacific Northwest USA. In the study by Hussain et al. (2018) the three top-ranked GCMs for the development of an MME for precipitation over Bornean tropical rainforests in Malaysia were used.

In the present study, the ensemble of GCMs was identified in two steps: (1) RM values of GCMs for annual, monsoon, winter, pre-monsoon and post-monsoon $P$, $T_{max}$ and $T_{min}$ were individually used to derive an overall rank for each GCM, and (2) four top-ranked GCMs based on RM values for all climate variables were considered for the ensemble. The selection of an appropriate set of GCMs considering their skills in different seasons enables the selection of an ensemble which can better simulate the observations in different seasons.

### 3.5 Development of Multi-model Ensemble Mean

The uncertainties in projections of a climate variable can be reduced by using its mean time series calculated from an MME of better performing GCMs (You et al., 2018). Numerous approaches are documented in the literature for the calculation of mean time series from an ensemble of better performing GCMs starting from simple arithmetic mean to machine learning algorithms (Kim et al., 2015). In the present study, two approaches 1). Simple Mean (SM) and 2). Random Forest (RF) (Breiman, 2001) were used for the calculation of mean time series of $P$, $T_{max}$ and $T_{min}$ corresponding to an ensemble of four top-ranked GCMs.

### 3.5.1 Simple Mean (SM)

Simple Mean (SM)-based MMEs were developed by simply averaging the individual $P$, $T_{max}$ and $T_{min}$ simulations of the four top-ranked GCMs using Eq.16.

$$SM = \frac{1}{n}\sum_{i=1}^{n} GCM_i \tag{16}$$

In Eq. 16, $n$ refers to the number of GCMs considered for the development of MMEs which is four in the present study and $GCM_i$ refers to the simulations of the climate variable of interest (i.e. $P$, $T_{max}$ and $T_{min}$) produced by the $i^{th}$ GCM.

### 3.5.2 Random Forest (RF)

Random Forest (RF) algorithm (Breiman, 2001) was used in the calculation of the mean time series of $P$, $T_{max}$ and $T_{min}$ corresponding to an MME of four top-ranked GCMs. RF is a relatively new machine learning algorithm widely used in modelling non-linear relationships between predictors and predictands (Ahmed et al., 2019b). RF algorithm is found to perform well with spatial data sets and less prone to over-fitting (Folberth et al., 2019). Most importantly Folberth et al. (2019) reported that RF is less sensitive to multivariate correlation. RF is an ensemble technique where regression is done using multiple decision trees. RF algorithm uses the following steps in developing regression models.

1. A bootstrap resampling method is used to select sample sets from training data (i.e. GCM and observed data).
2. Classification And Regression Tree (CART) technique is used to develop unpruned trees using the bootstrapped samples.
3. A large number of trees are developed with the samples selected repetitively from training data so that all training data have an equal probability of selection.
4. A regression model is fitted to each tree and the performance of each tree is assessed.
5. Ensemble simulation is estimated by averaging the predictions of all trees which is considered as the final simulation.

Wang et al. (2018) and He et al. (2016) reported that the performance of RF varies with the number of trees (*ntree*) and the number of variables randomly sampled (*mtry*) at each split in developing the trees. In those studies, it was observed that RF performance increases with the increase in the value of *ntree*. However, in the current study the performance was not found to increase significantly in term of root mean square error when the value of *ntree* was greater than 500. Therefore, *ntree* was set to 500 while the *mtry* was set to $p/3$ where $p$ is the number of variables (i.e. 4 GCMs) used for developing RF-based MME.

The MME prediction can be improved by assigning larger weights to the GCMs which show better performance (Sa'adi et al., 2017). RF regression models developed using historical $P$, $T_{max}$ and $T_{min}$ simulations of GCMs as independent variables and historical observed $P$, $T_{max}$ and $T_{min}$ as dependent variables provide weights to the GCMs according to their ability to simulate historical observed $P$, $T_{max}$ and $T_{min}$. The "randomForest" package (Breiman, 2006) written in R programming language was employed in this study for developing RF-based MMEs. RF-based MMEs were calibrated with the first 70% of the data and validated with the rest of the data.

# 4 Results and Discussion

## 4.1 Accuracy Assessment of Gridded Precipitation Data

As a preliminary analysis, the monthly time series of GPCC $P$, CRU $T_{max}$ and CRU $T_{min}$ data were validated against the monthly time series of observed $P$, $T_{max}$ and $T_{min}$. The validation was performed for the period 1961-2005. In the present study, two statistical metrics; Normalized Root Mean Square Error (NRMSE), and modified index of agreement ($md$) were used to assess the accuracy of monthly time series of GPCC $P$, CRU $T_{max}$ and CRU $T_{min}$ in replicating the mean and the variability of monthly time series of observed $P$, $T_{max}$ and $T_{min}$.

The NRMSE and $md$ values between observed $P$ and GPCC $P$ (pertaining to the grid point closest to the observation station), observed $T_{max}$ and $T_{min}$ with CRU $T_{max}$ and $T_{min}$ obtained for 17 locations in Pakistan are given in Table 2. Overall, all the stations showed low and high NRMSE and $md$ values respectively, indicating that the accuracy of the GPCC $P$ in replicating observed precipitation and CRU $T_{max}$ and CRU $T_{min}$ in replicating observed $T_{max}$ and $T_{min}$ over Pakistan is high. Overall, NRMSE values were found in the ranges of 0.09 to 0.970 for $P$, 0.100 to 0.390 for $T_{max}$, and 0.09 to 0.470 for $T_{min}$. Overall, $md$ values were found in the ranges of 0.680 to 0.960 for $P$, 0.810 to 0.960 for $T_{max}$, and 0.779 to 0.959 for $T_{min}$.

**Table 2.** Validation of accuracy of GPCC $P$ and CRU $T_{max}$ and $T_{min}$ using NRMSE and $md$

| Station No | Station Name | Precipitation ($P$) | | Maximum Temperature ($T_{max}$) | | Minimum Temperature ($T_{min}$) | |
|---|---|---|---|---|---|---|---|
| | | NRMSE | $md$ | NRMSE | $md$ | NRMSE | $md$ |
| 1 | Karachi | 0.530 | 0.840 | 0.270 | 0.880 | 0.180 | 0.919 |
| 2 | Pasni | 0.470 | 0.890 | 0.310 | 0.840 | 0.260 | 0.879 |
| 3 | Nawabshah | 0.740 | 0.740 | 0.300 | 0.850 | 0.170 | 0.919 |
| 4 | Padidan | 0.590 | 0.780 | 0.190 | 0.920 | 0.150 | 0.939 |
| 5 | Jacobabad | 0.520 | 0.840 | 0.100 | 0.960 | 0.090 | 0.959 |
| 6 | Dalbandin | 0.090 | 0.960 | 0.140 | 0.940 | 0.230 | 0.889 |
| 7 | Kalat | 0.970 | 0.870 | 0.240 | 0.900 | 0.470 | 0.779 |
| 8 | Sibbi | 0.590 | 0.880 | 0.390 | 0.810 | 0.260 | 0.889 |
| 9 | Bahawalnagar | 0.530 | 0.810 | 0.310 | 0.899 | 0.270 | 0.881 |
| 10 | Quetta | 0.750 | 0.760 | 0.240 | 0.890 | 0.120 | 0.949 |
| 11 | Multan | 0.730 | 0.740 | 0.120 | 0.950 | 0.120 | 0.949 |
| 12 | Faisalabad | 0.700 | 0.740 | 0.210 | 0.900 | 0.170 | 0.919 |
| 13 | Lahore | 0.710 | 0.700 | 0.140 | 0.940 | 0.110 | 0.959 |
| 14 | Sargodha | 0.790 | 0.680 | 0.160 | 0.930 | 0.170 | 0.919 |
| 15 | Mianwali | 0.720 | 0.750 | 0.240 | 0.890 | 0.120 | 0.949 |
| 16 | Islamabad | 0.450 | 0.840 | 0.160 | 0.930 | 0.190 | 0.909 |
| 17 | Peshawar | 0.690 | 0.720 | 0.190 | 0.920 | 0.110 | 0.949 |

## 4.2 Evaluation and Ranking of GCMs

SPAEF, FSS, Lambda, Cramer-V, Mapcurves, and KGE between observed (GPCC $P$, CRU $T_{max}$ and $T_{min}$) and GCM simulated mean annual, monsoon, winter, pre-monsoon and post-monsoon $P$, $T_{max}$ and $T_{min}$ of Pakistan were estimated for the period 1961 to 2005. As an example, Table 3 shows the GOF values that depict the performance of each GCM in simulating GPCC mean annual precipitation. In Table 3, the ranks of GCMs corresponding to each performance metric is shown within brackets. GOF values near to 1 refer to the better performance of the GCM of interest. For example, CESM1-CAM5 has a GOF value of 0.540 for SPAEF, and hence regarded as the best GCM in term of SPAEF, whereas CSIRO-Mk3-6-0 can be regarded as the poorest GCM which has a GOF value of -0.505 in term of SPAEF. The GOF values for other metrics (i.e. FSS, Lambda, Cramer-V, Mapcurves, and KGE) can also be interpreted in the same manner.

**Table 3.** GOF values and ranks of GCMs obtained using different spatial metrics for mean annual precipitation.

| GCM | SPAEF (Rank) | FSS (Rank) | Lambda (Rank) | Cramer-V(Rank) | Mapcurves (Rank) | KGE (Rank) |
|---|---|---|---|---|---|---|
| ACCESS1-0 | 0.411 (**7**) | 0.659 (**24**) | 0.143 (**24**) | 0.370 (**28**) | 0.244 (**29**) | 0.172 (**29**) |
| ACCESS1-3 | 0.155 (**24**) | 0.712 (**20**) | 0.107 (**30**) | 0.315 (**34**) | 0.206 (**34**) | 0.310 (**15**) |
| BCC-CSM1-1 | 0.241 (**21**) | 0.691 (**21**) | 0.143 (**24**) | 0.388 (**27**) | 0.258 (**27**) | 0.082 (**33**) |
| BCC-CSM1.1(m) | 0.149 (**25**) | 0.685 (**22**) | 0.214 (**13**) | 0.545 (**16**) | 0.376 (**16**) | 0.304 (**16**) |
| BNU-ESM | 0.185 (**23**) | 0.759 (**11**) | 0.179 (**18**) | 0.519 (**21**) | 0.349 (**21**) | 0.233 (**26**) |
| CanESM2 | 0.250 (**20**) | 0.642 (**26**) | 0.250 (**6**) | 0.547 (**15**) | 0.378 (**15**) | -0.443 (**35**) |
| CCSM4 | 0.440 (**4**) | 0.798 (**5**) | 0.250 (**6**) | 0.667 (**4**) | 0.525 (**4**) | 0.420 (**8**) |
| CESM1-BGC | 0.439 (**5**) | 0.759 (**12**) | 0.214 (**13**) | 0.655 (**10**) | 0.508 (**10**) | 0.337 (**12**) |
| CESM1-CAM5 | 0.540 (**1**) | 0.840 (**1**) | 0.250 (**6**) | 0.667 (**4**) | 0.525 (**4**) | 0.531 (**2**) |
| CESM1-WACCM | 0.430 (**6**) | 0.776 (**10**) | 0.250 (**6**) | 0.656 (**9**) | 0.510 (**9**) | 0.384 (**10**) |
| CMCC-CM | -0.255 (**34**) | 0.565 (**33**) | 0.143 (**24**) | 0.496 (**24**) | 0.325 (**24**) | 0.189 (**28**) |
| CMCC-CMS | -0.043 (**28**) | 0.637 (**28**) | 0.143 (**24**) | 0.369 (**29**) | 0.244 (**28**) | 0.249 (**22**) |
| CNRM-CM5 | 0.364 (**12**) | 0.732 (**17**) | 0.250 (**6**) | 0.667 (**4**) | 0.525 (**4**) | 0.314 (**14**) |
| CSIRO-Mk3-6-0 | -0.505 (**36**) | 0.321 (**36**) | 0.036 (**36**) | 0.264 (**36**) | 0.179 (**36**) | -1.837 (**36**) |
| EC-EARTH | 0.232 (**22**) | 0.756 (**13**) | 0.286 (**4**) | 0.759 (**2**) | 0.642 (**2**) | 0.404 (**9**) |
| FGOALS-g2 | 0.321 (**13**) | 0.793 (**6**) | 0.179 (**18**) | 0.531 (**17**) | 0.361 (**17**) | 0.362 (**11**) |
| FIO-ESM | 0.281 (**17**) | 0.752 (**14**) | 0.214 (**13**) | 0.559 (**14**) | 0.391 (**14**) | 0.283 (**19**) |
| GFDL-CM3 | 0.387 (**8**) | 0.815 (**4**) | 0.429 (**1**) | 0.782 (**1**) | 0.690 (**1**) | 0.493 (**3**) |
| GFDL-ESM2G | 0.307 (**14**) | 0.786 (**7**) | 0.250 (**6**) | 0.667 (**4**) | 0.525 (**4**) | 0.484 (**4**) |
| GFDL-ESM2M | 0.297 (**16**) | 0.778 (**8**) | 0.214 (**13**) | 0.436 (**26**) | 0.296 (**25**) | 0.458 (**5**) |
| GISS-E2-H | -0.100 (**32**) | 0.616 (**31**) | 0.107 (**30**) | 0.335 (**33**) | 0.220 (**33**) | 0.245 (**24**) |
| GISS-E2-R | -0.054 (**29**) | 0.616 (**30**) | 0.107 (**30**) | 0.350 (**31**) | 0.229 (**31**) | 0.236 (**25**) |

| | | | | | |
|---|---|---|---|---|---|
| HadGEM2-AO | 0.454 (**3**) | 0.740 (**15**) | 0.179 (**18**) | 0.520 (**20**) | 0.350 (**20**) | 0.315 (**13**) |
| HadGEM2-CC | 0.387 (**9**) | 0.683(**23**) | 0.179 (**18**) | 0.360 (**30**) | 0.236 (**30**) | 0.222 (**27**) |
| HadGEM2-ES | 0.371 (**11**) | 0.721 (**18**) | 0.179 (**18**) | 0.530 (**18**) | 0.360 (**18**) | 0.277 (**20**) |
| INMCM4 | 0.378 (**10**) | 0.777 (**9**) | 0.179 (**18**) | 0.530 (**18**) | 0.360 (**18**) | 0.422 (**6**) |
| IPSL-CM5A-LR | -0.054 (**30**) | 0.634 (**29**) | 0.357 (**2**) | 0.590 (**12**) | 0.427 (**12**) | 0.117 (**32**) |
| IPSL-CM5A-MR | -0.093 (**31**) | 0.548 (**34**) | 0.357 (**2**) | 0.590 (**12**) | 0.427 (**12**) | -0.183 (**34**) |
| IPSL-CM5B-LR | -0.286 (**35**) | 0.538(**35**) | 0.107 (**30**) | 0.350 (**31**) | 0.229 (**31**) | 0.131 (**31**) |
| MIROC-ESM-CHEM | 0.273 (**18**) | 0.733(**16**) | 0.214 (**13**) | 0.655 (**10**) | 0.508 (**10**) | 0.303 (**17**) |
| MIROC-ESM | 0.258 (**19**) | 0.720 (**19**) | 0.286 (**4**) | 0.677 (**3**) | 0.537 (**3**) | 0.290 (**18**) |
| MIROC5 | 0.302 (**15**) | 0.828 (**3**) | 0.071 (**34**) | 0.454 (**25**) | 0.285 (**26**) | 0.420 (**7**) |
| MPI-ESM-LR | -0.012 (**27**) | 0.639 (**27**) | 0.143 (**24**) | 0.517 (**22**) | 0.346 (**22**) | 0.253 (**21**) |
| MPI-ESM-MR | 0.041 (**26**) | 0.653 (**25**) | 0.143 (**24**) | 0.506 (**23**) | 0.335 (**23**) | 0.245 (**23**) |
| MRI-CGCM3 | -0.180 (**33**) | 0.572 (**32**) | 0.071 (**34**) | 0.293 (**35**) | 0.194 (**35**) | 0.169 (**30**) |
| NorESM1-M | 0.464 (**2**) | 0.833 (**2**) | 0.250 (**6**) | 0.667 (**4**) | 0.525 (**4**) | 0.532 (**1**) |

Table 3 shows the ranks attained by GCMs corresponding to different metrics. For example, BCC-CSM1.1 (m) attained ranks 25, 22, 13, 16, 16 and 16 in terms of SPAEF, FSS, Lambda, Cramer-V, Mapcurves, and KGE respectively. It was observed that CSIRO-Mk3-6-0 is the only GCM which was able to secure the same rank for all metrics. However, HadGEM2-ES secured rank 18 for four metrics (i.e. FSS, Lambda, Cramer-V, Mapcurves). Several GCMs attained the same rank for three metrics (e.g. BCC-CSM1.1(m), CCSM4, CMCC-CM and CMCC-CMS). Cramer-V and Mapcurve showed more or less similar ranks for GCMs. Similar results were also seen for other seasons and variables (not presented in the manuscript).

## 4.3 Overall Ranks of GCMs for Precipitation, Maximum Temperature and Minimum Temperature

The application of various evaluation metrics has yielded different ranks for the same GCM (Ahmadalipour et al., 2017;Raju et al., 2017). The ranks attained by GCMs corresponding to different metrics and seasons (annual, monsoon, winter, pre-monsoon and post-monsoon) were used to calculate the RM values for each GCM. The ranks of GCMs for $P$, $T_{max}$ and $T_{min}$ are presented in Table 4 along with the RM values. As seen in Table 4, EC-EARTH, BCC-CSM1.1 (m) and CSIRO-Mk3-6-0 were the most skilful GCMs in reproducing the spatial characteristics of $P$, $T_{max}$ and $T_{min}$ respectively. On the other hand, IPSL-CM5B-LR, CMCC-CM, and INMCM4 were poorest GCMs in reproducing the spatial characteristics of $P$, $T_{max}$ and $T_{min}$ respectively.

The better performance of EC-EARTH, BCC-CSM1.1 (m) and CSIRO-Mk3-6-0 in simulating $P$, $T_{max}$ and $T_{min}$ over Indo-Pak sub-continent has also been reported in several past studies. Latif et al. (2018) reported the relatively better performance of EC-EARTH, and BCC-CSM1.1 (m) out of 36 CMIP5 GCMs in simulating precipitation over Indo-Pakistan sub-continent based on spatial correlations. Rehman et al. (2018) conducted a study to assess the performance of CMIP5 GCMs in

simulating mean precipitation and temperature over south Asia. The study reported the better performance of EC-EARTH in simulating precipitation and CSIRO-Mk3-6-0 in simulating temperature. Khan et al. (2018a) assessed the performance of 31 CMIP5 GCMs in simulating mean precipitation and temperature over Pakistan using multiple daily gridded datasets and identified EC-EARTH as the best GCM for simulating precipitation and CSIRO-Mk3-6-0 for simulating temperature. Better performance of CSIRO-Mk3-6-0 in simulating maximum and minimum temperature is also reported in the study by (Ahmed et al., 2019c).

**Table 4.** Ranks of GCMs for $P$, $T_{max}$ and $T_{min}$ based on rating metric values

| GCM | $P$ | Rank | GCM | $T_{max}$ | Rank | GCM | $T_{min}$ | Rank |
|---|---|---|---|---|---|---|---|---|
| EC-EARTH | 0.823 | 1 | BCC-CSM1.1(m) | 0.702 | 1 | CSIRO-Mk3-6-0 | 0.750 | 1 |
| NorESM1-M | 0.794 | 2 | NorESM1-M | 0.663 | 2 | GFDL-ESM2G | 0.720 | 2 |
| GFDL-CM3 | 0.714 | 3 | HadGEM2-ES | 0.656 | 3 | CMCC-CMS | 0.692 | 3 |
| CCSM4 | 0.689 | 4 | IPSL-CM5B-LR | 0.630 | 4 | BCC-CSM1.1(m) | 0.684 | 4 |
| MIROC5 | 0.685 | 5 | HadGEM2-AO | 0.626 | 5 | GFDL-ESM2M | 0.681 | 5 |
| GFDL-ESM2G | 0.673 | 6 | CMCC-CMS | 0.616 | 6 | MIROC-ESM-CHEM | 0.657 | 6 |
| CESM1-CAM5 | 0.654 | 7 | HadGEM2-CC | 0.608 | 7 | NorESM1-M | 0.656 | 7 |
| HadGEM2-AO | 0.651 | 8 | FGOALS-g2 | 0.600 | 8 | ACCESS1-3 | 0.656 | 8 |
| GFDL-ESM2M | 0.643 | 9 | CSIRO-Mk3-6-0 | 0.594 | 9 | MIROC-ESM | 0.654 | 9 |
| FGOALS-g2 | 0.607 | 10 | ACCESS1-0 | 0.577 | 10 | MIROC5 | 0.646 | 10 |
| MIROC-ESM | 0.589 | 11 | IPSL-CM5A-LR | 0.566 | 11 | CCSM4 | 0.631 | 11 |
| ACCESS1-0 | 0.555 | 12 | INMCM4 | 0.561 | 12 | CESM1-BGC | 0.628 | 12 |
| ACCESS1-3 | 0.555 | 12 | GISS-E2-H | 0.556 | 13 | CESM1-CAM5 | 0.595 | 13 |
| MIROC-ESM-CHEM | 0.532 | 14 | MIROC5 | 0.551 | 14 | MRI-CGCM3 | 0.584 | 14 |
| HadGEM2-CC | 0.531 | 15 | BNU-ESM | 0.538 | 15 | CanESM2 | 0.577 | 15 |
| HadGEM2-ES | 0.514 | 16 | BCC-CSM1-1 | 0.534 | 16 | BNU-ESM | 0.569 | 16 |
| BCC-CSM1-1 | 0.506 | 17 | GISS-E2-R | 0.532 | 17 | FGOALS-g2 | 0.569 | 16 |
| CESM1-WACCM | 0.482 | 18 | MPI-ESM-LR | 0.532 | 17 | MPI-ESM-MR | 0.569 | 16 |
| CNRM-CM5 | 0.480 | 19 | FIO-ESM | 0.524 | 19 | MPI-ESM-LR | 0.566 | 19 |
| CESM1-BGC | 0.467 | 20 | CESM1-WACCM | 0.522 | 20 | EC-EARTH | 0.506 | 20 |
| INMCM4 | 0.464 | 21 | ACCESS1-3 | 0.520 | 21 | IPSL-CM5A-MR | 0.490 | 21 |
| FIO-ESM | 0.462 | 22 | GFDL-ESM2M | 0.514 | 22 | HadGEM2-ES | 0.487 | 22 |
| MPI-ESM-MR | 0.437 | 23 | MPI-ESM-MR | 0.513 | 23 | ACCESS1-0 | 0.481 | 23 |
| IPSL-CM5A-LR | 0.426 | 24 | CCSM4 | 0.466 | 24 | FIO-ESM | 0.446 | 24 |
| CanESM2 | 0.406 | 25 | CESM1-BGC | 0.459 | 25 | CMCC-CM | 0.428 | 25 |
| MPI-ESM-LR | 0.395 | 26 | CanESM2 | 0.442 | 26 | GISS-E2-R | 0.418 | 26 |
| BCC-CSM1.1(m) | 0.394 | 27 | MIROC-ESM | 0.442 | 26 | GISS-E2-H | 0.416 | 27 |
| IPSL-CM5A-MR | 0.382 | 28 | CNRM-CM5 | 0.434 | 28 | HadGEM2-AO | 0.416 | 27 |

| | | | | | | | | |
|---|---|---|---|---|---|---|---|---|
| CMCC-CMS | 0.381 | 29 | EC-EARTH | 0.427 | 29 | IPSL-CM5A-LR | 0.416 | 27 |
| MRI-CGCM3 | 0.381 | 29 | MIROC-ESM-CHEM | 0.427 | 29 | BCC-CSM1-1 | 0.413 | 30 |
| CMCC-CM | 0.353 | 31 | GFDL-ESM2G | 0.416 | 31 | HadGEM2-CC | 0.413 | 30 |
| BNU-ESM | 0.337 | 32 | GFDL-CM3 | 0.398 | 32 | CNRM-CM5 | 0.361 | 32 |
| GISS-E2-H | 0.319 | 33 | CESM1-CAM5 | 0.371 | 33 | CESM1-WACCM | 0.356 | 33 |
| CSIRO-Mk3-6-0 | 0.273 | 34 | IPSL-CM5A-MR | 0.326 | 34 | IPSL-CM5B-LR | 0.275 | 34 |
| GISS-E2-R | 0.253 | 35 | MRI-CGCM3 | 0.319 | 35 | GFDL-CM3 | 0.231 | 35 |
| IPSL-CM5B-LR | 0.144 | 36 | CMCC-CM | 0.249 | 36 | INMCM4 | 0.226 | 36 |

The spatial patterns of mean annual $P$, $T_{max}$ and $T_{min}$ simulated by the GCMs ranked 1 and 36 were compared with the spatial patterns of GPCC $P$ and CRU $T_{max}$ and $T_{min}$ and presented in Figure 3 as an example. In Figure 3 it was seen that the GCMs that attained rank 1 (the best performing GCM) showed spatial patterns more or less similar to that of GPCC $P$ and CRU $T_{max}$ and $T_{min}$. On the other hand, GCMs ranked 36 (the worst-performing GCM) showed large differences compared to the spatial patterns of GPCC $P$ and CRU $T_{max}$ and $T_{min}$. Figure 3 clearly shows that GCMs which attained rank 36 under-estimated the precipitation and temperature over a large region in the study area.

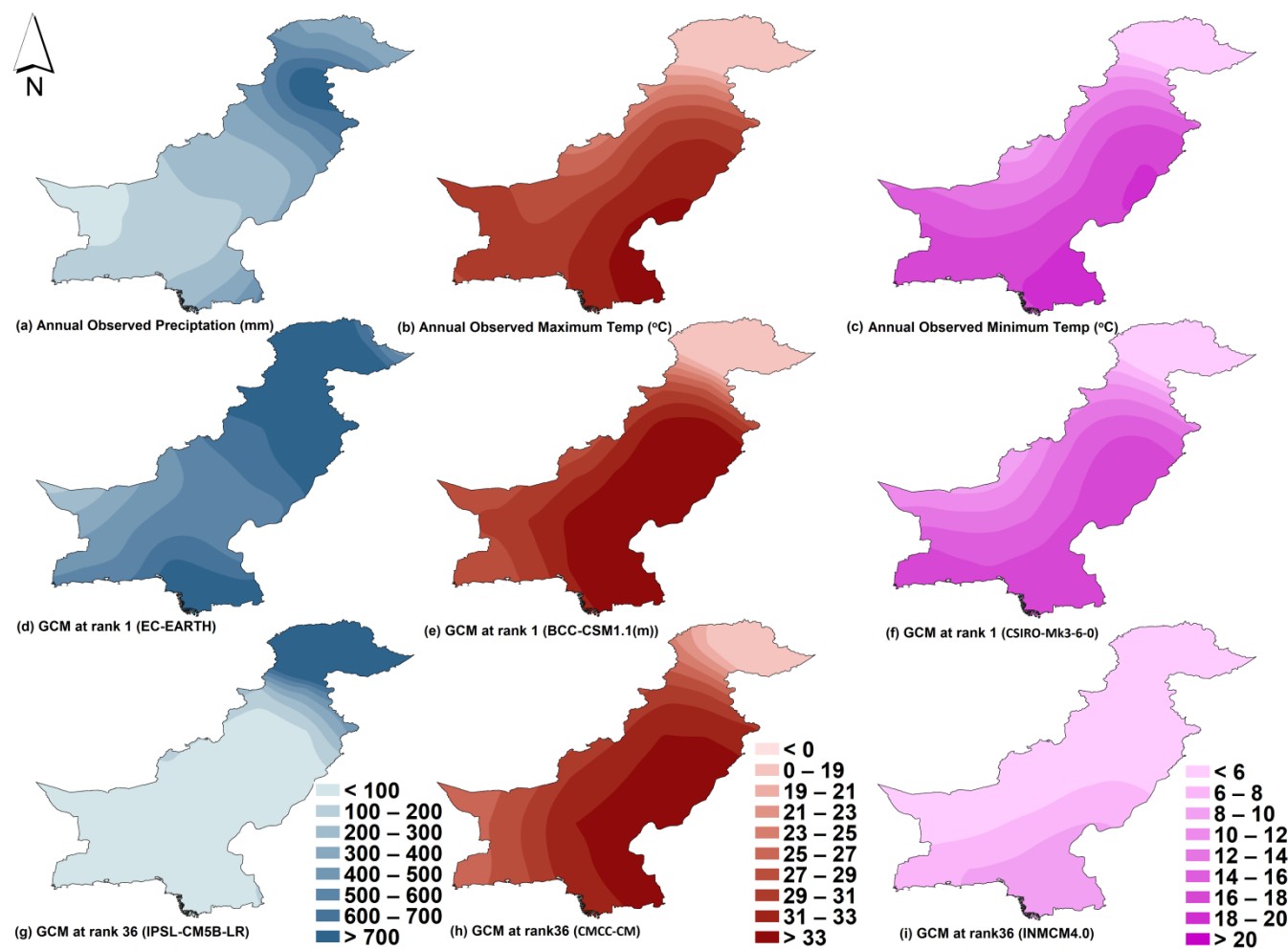

**Figure 3.** Spatial patterns of (a) GPCC precipitation, (b) CRU maximum temperature, (c) CRU minimum temperature, (d - f) GCM ranked 1 and (g - i) GCM ranked 36 for mean annual precipitation, maximum and minimum temperature for the period 1961 to 2005.

## 4.4 Identification of Ensemble Members

Based on the criteria mentioned in Section 3.4, ranks of each variable were estimated and then the GCMs were ranked based on the overall RM values. Table 5 shows the overall ranks of the 36 GCMs considered in this study. The four top-ranked GCMs; NorESM1-M, MIROC5, BCC-CSM1-1 and ACCESS1-3 that are indicated in bold text in Table 5 were selected as the members of the ensemble for $P$, $T_{max}$ and $T_{min}$ over Pakistan.

**Table 5.** Overall ranks of GCMs for the identification of ensemble members

| GCM | $P$ Rank | $T_{max}$ Rank | $T_{min}$ Rank | Overall RM Value | Overall Rank |
|---|---|---|---|---|---|
| **NorESM1-M** | 2 | 2 | 7 | 0.898 | **1** |
| **MIROC5** | 5 | 14 | 10 | 0.731 | **2** |
| **BCC-CSM1-1** | 17 | 16 | 30 | 0.417 | **3** |
| **ACCESS1-3** | 10 | 8 | 16 | 0.685 | **4** |
| GFDL-ESM2M | 9 | 22 | 5 | 0.667 | 5 |
| CMCC-CMS | 29 | 6 | 3 | 0.648 | 6 |
| CCSM4 | 4 | 24 | 11 | 0.639 | 7 |
| GFDL-ESM2G | 6 | 31 | 2 | 0.639 | 8 |
| HadGEM2-AO | 8 | 5 | 27 | 0.630 | 9 |
| FGOALS-g2 | 12 | 21 | 8 | 0.620 | 10 |
| HadGEM2-ES | 16 | 3 | 22 | 0.620 | 11 |
| CSIRO-Mk3-6-0 | 34 | 9 | 1 | 0.593 | 12 |
| ACCESS1-0 | 12 | 10 | 23 | 0.583 | 13 |
| MIROC-ESM-CHEM | 14 | 29 | 6 | 0.546 | 14 |
| MIROC-ESM | 11 | 26 | 9 | 0.574 | 15 |
| EC-EARTH | 1 | 29 | 20 | 0.537 | 16 |
| HadGEM2-CC | 15 | 7 | 30 | 0.519 | 17 |
| CESM1-CAM5 | 7 | 33 | 13 | 0.509 | 18 |
| CESM1-BGC | 20 | 25 | 12 | 0.472 | 19 |
| IPSL-CM5A-LR | 24 | 11 | 27 | 0.426 | 20 |
| MPI-ESM-LR | 26 | 17 | 19 | 0.426 | 21 |
| MPI-ESM-MR | 23 | 23 | 16 | 0.426 | 22 |
| BCC-CSM1.1(m) | 27 | 1 | 4 | 0.704 | 23 |
| BNU-ESM | 32 | 15 | 16 | 0.417 | 24 |
| FIO-ESM | 22 | 19 | 24 | 0.398 | 25 |
| CanESM2 | 25 | 26 | 15 | 0.389 | 26 |
| INMCM4 | 21 | 12 | 36 | 0.361 | 27 |
| GFDL-CM3 | 3 | 32 | 35 | 0.352 | 28 |
| CESM1-WACCM | 18 | 20 | 33 | 0.343 | 29 |
| GISS-E2-H | 33 | 13 | 27 | 0.324 | 30 |
| IPSL-CM5B-LR | 36 | 4 | 34 | 0.315 | 31 |
| GISS-E2-R | 35 | 17 | 26 | 0.278 | 32 |
| MRI-CGCM3 | 29 | 35 | 14 | 0.278 | 33 |
| CNRM-CM5 | 19 | 28 | 32 | 0.269 | 34 |
| IPSL-CM5A-MR | 28 | 34 | 21 | 0.231 | 35 |
| CMCC-CM | 31 | 36 | 25 | 0.148 | 36 |

The performances of the four top-ranked GCMs (i.e. GCMs ranked 1, 2, 3 and 4) and four lowest-ranked GCMs (i.e. GCMs ranked 33, 34, 35, and 36) were visually evaluated using scatter plots shown in Figures 4 and 5, pertaining to mean annual $P$, $T_{max}$ and $T_{min}$ as example. In order to plot the scatter, the $P$, $T_{max}$ and $T_{min}$ simulated by each GCM and GPCC $P$, CRU $T_{max}$ and CRU $T_{min}$ pertaining to all grid points was averaged (spatially averaged precipitation and temperature). As expected, GCMs that attained ranks 1 to 4 showed a close agreement with the GPCC $P$, CRU $T_{max}$ and CRU $T_{min}$ compared to that of GCMs which attained ranks 33, 34, 35, and 36. The same can also be noticed based on $md$ values provided in each figure where top-ranked GCMs showed higher $md$ values compared to the lowest-ranked GCMs. The scatter plots in Figure 5 indicated that the least skilful GCMs underestimated mean annual $P$, $T_{max}$ and $T_{min}$. Over and underestimation of $P$, $T_{max}$ and $T_{min}$ can also be seen in the scatter plots of GCMs ranked 1, 2, 3 and 4. However, their scatter was found much aligned with the 45-degree line compared to that of GCMs ranked 33, 34, 35, and 36. Therefore, it is argued that the GCMs ranked 1, 2, 3 and 4 can be used as an ensemble for the simulation of $P$, $T_{max}$ and $T_{min}$.

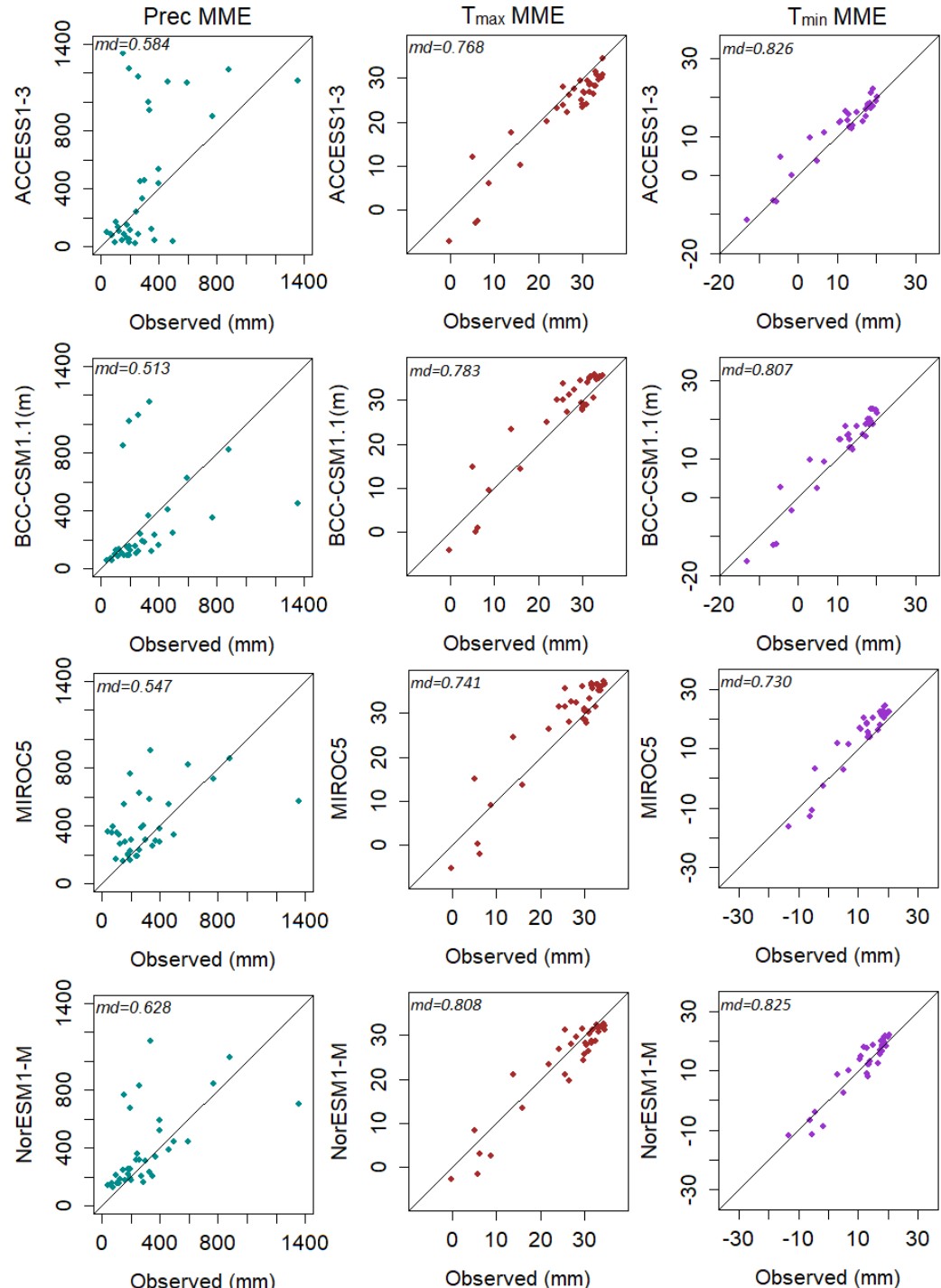

**Figure 4.** Scatter of spatially averaged annual *P*, *T_max* and *T_min* of four top-ranked GCMs plotted against GPCC *P*, CRU *T_max* and CRU *T_min* for the period 1961 to 2005.

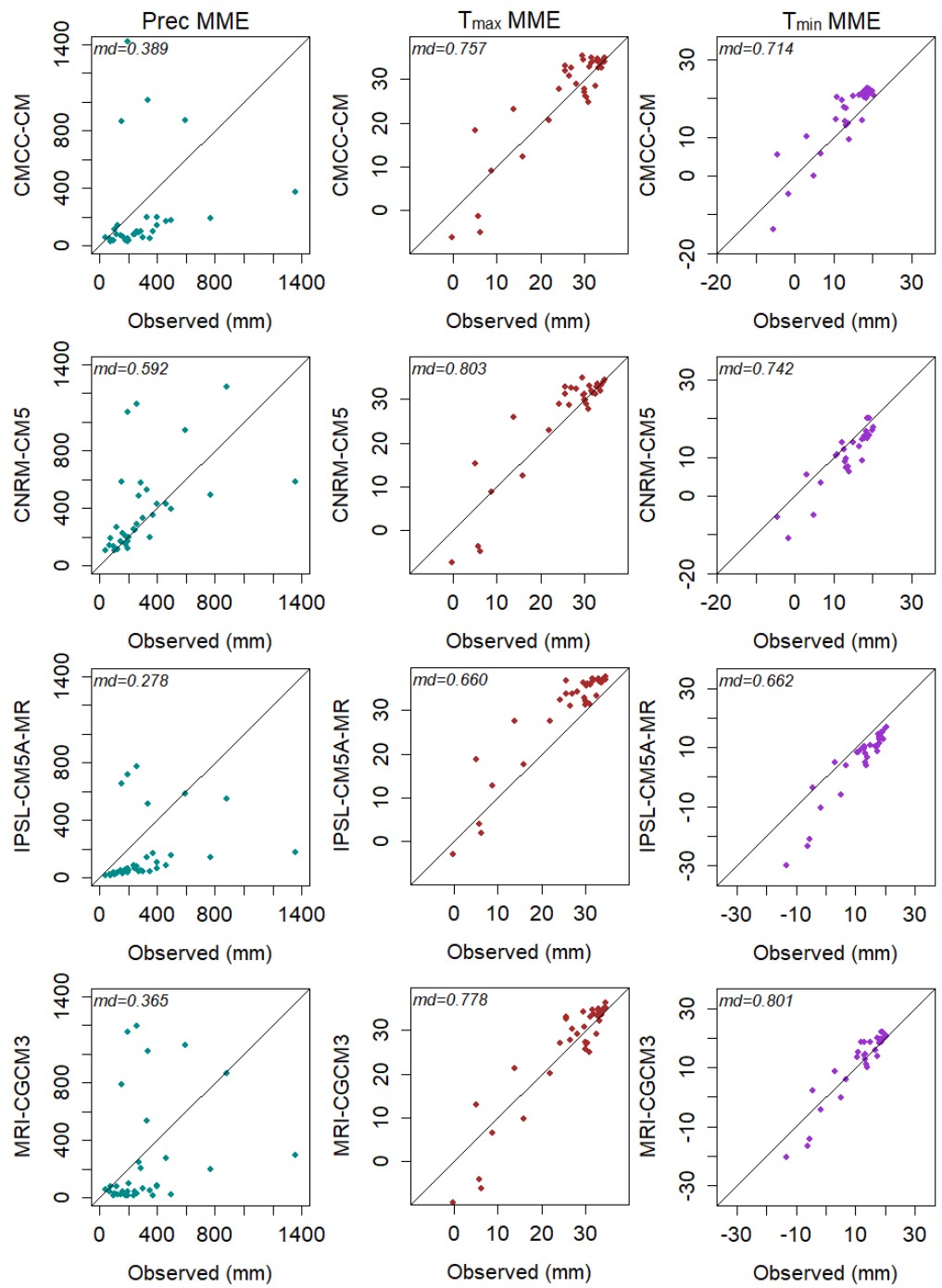

**Figure 5.** Scatter of spatially averaged annual *P*, $T_{max}$ and $T_{min}$ of four lowest ranked GCMs plotted against GPCC *P*, CRU $T_{max}$ and CRU $T_{min}$ for the period 1961 to 2005.

Some of the GCMs identified for the ensemble over Pakistan in this study have also been identified as better-performing GCMs over neighboring countries such as India and Iran. Jena et al. (2015) used Z-value test, correlation coefficient, relative precipitation comparison test, probability function comparison, root mean square error, and Student's t-test to evaluate the performance of 20 CMIP5 GCMs in simulating Indian summer monsoon. They found that CCSM4, CESM1-CAM5, GFDL-CM3, and GFDL-ESM2G perform better compared to the other GCMs. Prasanna (2015) conducted a study to assess the performance of 12 CMIP5 GCMs using mean and coefficient of variation over South Asia (5N–35N; 65E–95E) and identified ACCESS, CNRM, HadGEM2-ES, MIROC5, Can-ESM, GFDL-ESM2M, GISS, MPI-ESM and NOR-ESM as better-performing GCMs. Sarthi et al. (2016) evaluated the performance of 34 CMIP5 GCMs using Taylor diagram, skill score, correlation and RMSE. They found that BCC-CSM1.1(m), CCSM4, CESM1(BGC), CESM1(CAM5), CESM1(WACCM), and MPI-ESM-MR were able to better capture the Indian summer monsoon precipitation. Afshar et al. (2016) applied Nash–Sutcliffe efficiency, percent of bias, coefficient of determination, and the ratio of RMSE to standard deviation of observations for assessing the performance of precipitation simulations of 14 CMIP5 GCMs over a mountainous catchment in north-eastern Iran which borders Pakistan. They recommend GFDL-ESM2G, IPSL-CM5A-MR, MIROC-ESM, and NorESM1-M as better GCMs. Mahmood et al. (2018) used correlation coefficient, the error between observed and GCM mean and standard deviation, and root mean square error to assess the performance of CMIP5 GCMs in simulating precipitation over Jhelum river basin, Pakistan and reported the good performance of GFDL-ESM2G, HadGEM2-ES, NorESM1-ME, CanESM2, and MIROC5. Latif et al. (2018) reported better performance of HadGEM2-AO, INM-CM4, CNRM-CM5, NorESM1-M, CCSM4 and CESM1-WACCM out of 36 GCMs in simulating precipitation over Indo-Pakistan region based on partial correlation. The above findings indicated that the GCMs identified in this study for the ensemble were also found to perform well in the other studies conducted over nearby countries/regions.

## 4.5 Multi-model Ensemble (MME) Mean

The performance of GCM ensembles identified in Section 4.4 was validated considering two types of MME means. The MME mean of $P$, $T_{max}$ and $T_{min}$ of the four top-ranked GCMs was calculated with (1). Simple Mean (SM) and (2). Random Forest (RF). In the application of SM, the time series of $P$, $T_{max}$ and $T_{min}$ of the four top-ranked GCMs were averaged to obtain the MME while in the application of RF, the time series of $P$, $T_{max}$ and $T_{min}$ of the four top-ranked GCMs were considered as inputs to the RF-based MME.

In Figure 6, the spatial patterns of $P$, $T_{max}$ and $T_{min}$ corresponding to both MMEs derived with SM and RF were compared with those of GPCC $P$, CRU $T_{max}$ and CRU $T_{min}$. The spatial patterns of $P$, $T_{max}$ and $T_{min}$ were created using ordinary kriging technique. Ordinary kriging was selected as it was found to perform better than other interpolation methods over Pakistan (Ahmed et al., 2014). As seen in Figure 6, both MMEs captured the spatial patterns of observed $P$, $T_{max}$ and $T_{min}$ to a good degree. However, the differences can be seen in both MMEs in replicating the spatial pattern of GPCC $P$, CRU $T_{max}$ and CRU $T_{min}$. The visual comparison provided in Figure 6 also indicated that RF-based MME performs better than the MME based on SM. SM-based MME was found to underestimate annual precipitation in the south-western and the northern

regions, while the RF-based MME was found to produce a spatial pattern almost identical to that of GPCC precipitation. A similar result can also be seen for $T_{max}$ and $T_{min}$ patterns where RF-based MME showed better performance. The better performance of RF in generating MMEs has also been reported in several other studies. Salman et al. (2018a) generated MME mean for maximum and minimum temperature over Iraq using four CMIP5 GCMs and reported that RF-based MME

5   performed better compared to individual GCMs. Likewise, Wang et al. (2018) conducted a comprehensive study to evaluate the performance of different machine learning techniques including RF, support vector machine, Bayesian model averaging and the arithmetic ensemble mean in generating MMEs. They considered 33 CMIP5 GCMs for precipitation and temperature over 108 stations located in Australia and concluded that RF and SVM can generate better performing MMEs compared to other techniques.

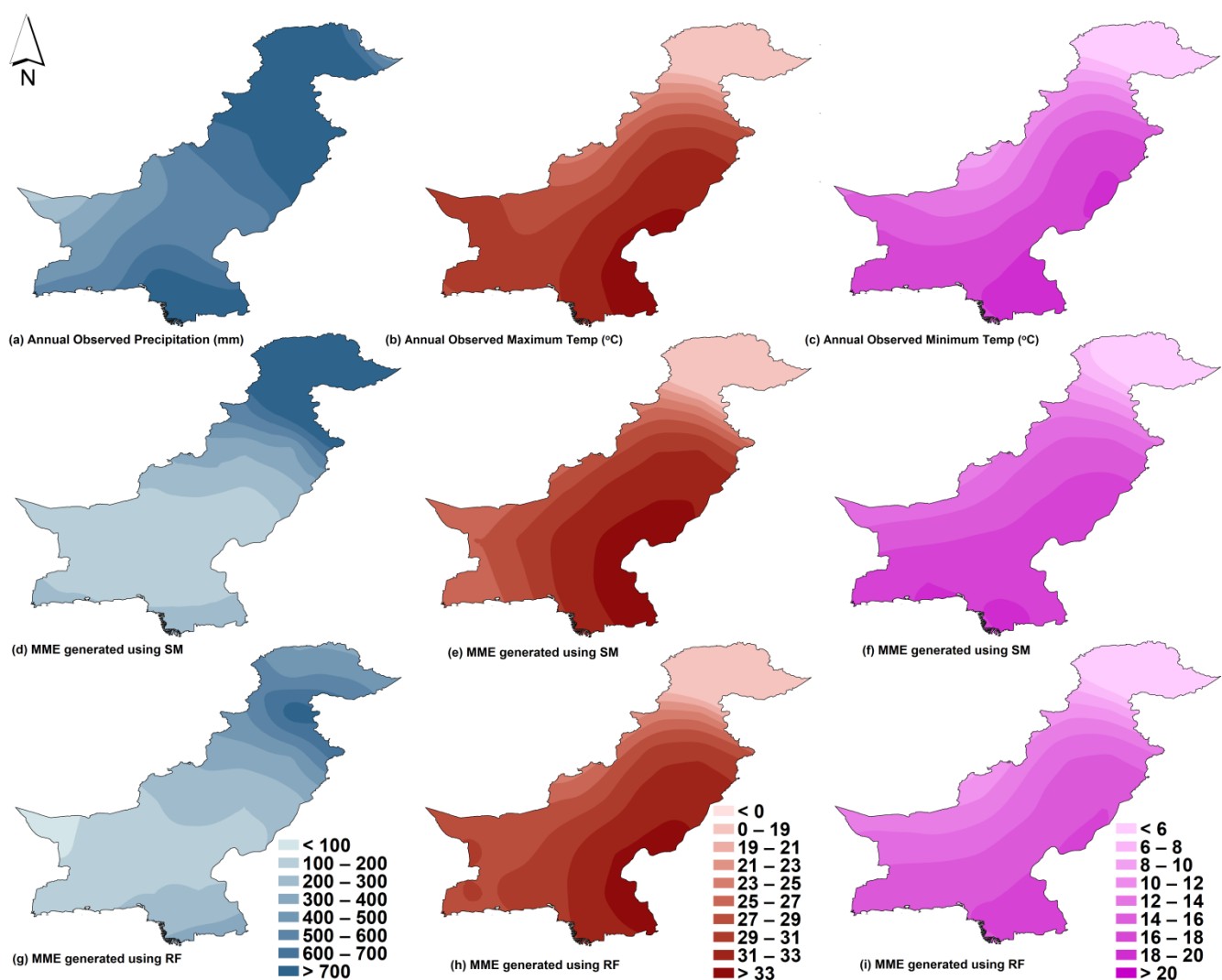

**Figure 6.** Spatial patterns of (a) GPCC precipitation, (b) CRU maximum temperature, (c) CRU minimum temperature, (d – f) MME-based on Simple Mean (SM) and (g - i) MME-based on Random Forest (RF) for mean annual precipitation, maximum and minimum temperature for the period 1961 to 2005.

The performance of MME ensembles was further evaluated using scatter plots shown in Figure 7. Scatter plots were developed using spatially averaged GPCC $P$, CRU $T_{max}$ and CRU $T_{min}$ and MME annual $P$, $T_{max}$ and $T_{min}$ at all grid points for the period 1961-2005. According to scatter plots in Figure 7, RF-based MME performed significantly better compared to its counterpart SM-based MME in simulating $P$, $T_{max}$ and $T_{min}$.

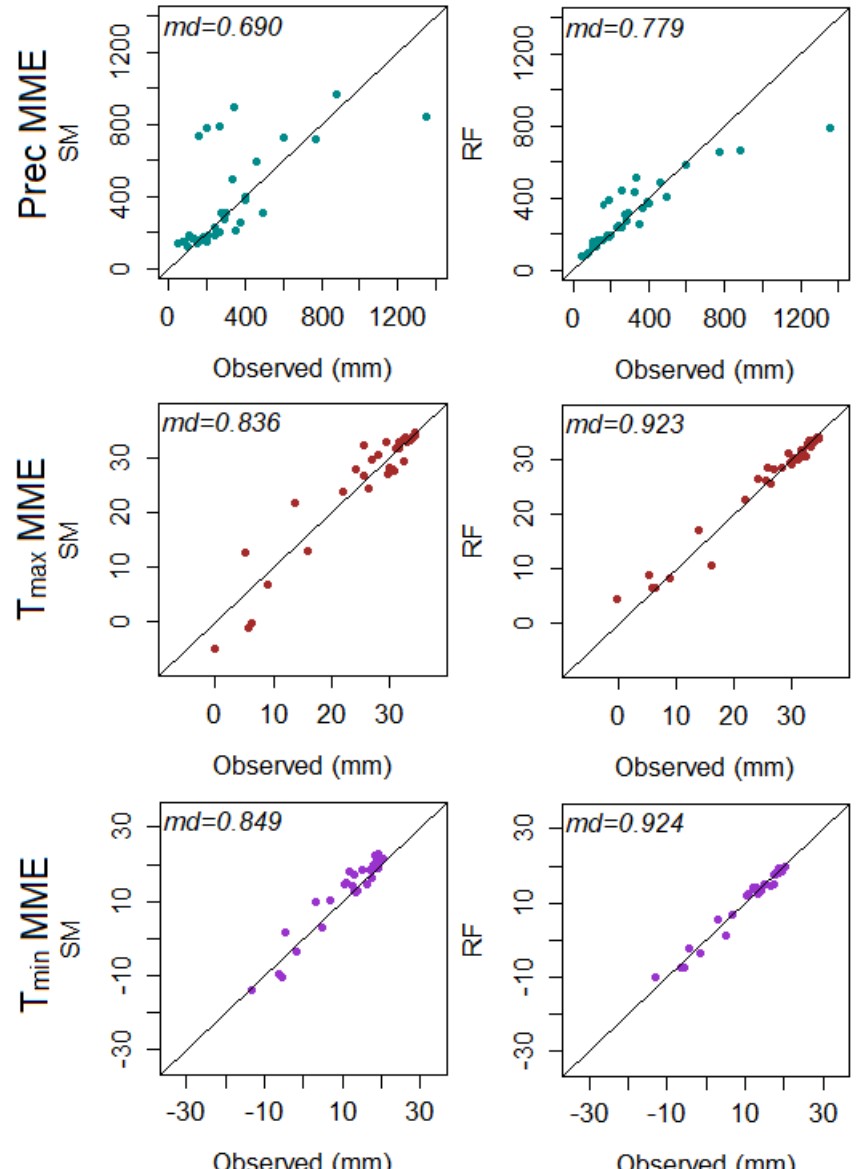

**Figure 7.** Scatter of spatially averaged annual $P$, $T_{max}$ and $T_{min}$ of MMEs developed with Simple Mean (SM) and Random Forest (RF) using four top-ranked GCMs plotted against GPCC $P$, CRU $T_{max}$ and CRU $T_{min}$ for the period 1961 to 2005.

5    In this study performance of GCMs was assessed based on their ability to simulate past observed $P$, $T_{max}$ and $T_{min}$ and hence the best performing GCMs were identified and used for the development of MMEs. However, it is found that past and future climate may have a weak association hence it is not guaranteed that a GCM performs well in the past will produce reliable results in future (Knutti et al., 2010). In other words, the best GCMs selected for the MMEs considering their ability to

simulate past climate may not be the best in the future under changing climate (Ruane and McDermid, 2017;Ahmed et al., 2019c). This is due to the large uncertainties associated with GHG emission scenarios and GCMs. As a solution to this limitation, Salman et al. (2018a) selected an ensemble of GCMs based on past performance as well as the degree of agreement between their future projections. The study detailed in the present manuscript can be repeated in future to select GCMs considering their past performance and the degree of agreement in their future projections.

In the present study, the MME of $P$, $T_{max}$ and $T_{min}$ were developed by considering four top-ranked GCMs. In the past, MMEs were developed considering 3 to 10 top-ranked GCMs. However, none of the past studies investigated the performance of MMEs by varying the number of GCMs used in developing them in MME. The performance of an MME can be sensitive to the choice of the number of GCMs. Hence, in future, a study should be conducted to investigate the impact of the number of GCMs used for the development of the MME.

Only RF algorithm was used in this study for the development of MMEs. Other machine learning algorithms (e.g. Artificial Neural Networks, Support Vector Machine, Relevance Vector Machine, K-nearest neigbour, Extreme Learning Machine) can also be used for the development of MMEs. A comparison of the performance of MMEs developed with different machine learning algorithms can assist in the identification of the pros and cons of different algorithms in relation to development of MMEs.

In the present study, GCM ranking and MME development was conducted only considering $P$, $T_{max}$ and $T_{min}$ pertaining to annual, monsoon, winter, pre-monsoon and post-monsoon seasons. However, several studies reported that the ranking of GCMs based on a variety climate variables may assist in the identification of a more dependable set of GCMs for an MME (Johnson and Sharma, 2012;Xuan et al., 2017). In future, the ranking of GCMs can be conducted considering a number of climate variables such as precipitation, mean temperature, maximum temperature, minimum temperature, wind speed, evapotranspiration and solar radiation.

## 5. Conclusions

This study quantitatively and qualitatively assessed the spatial accuracy of 36 CMIP5 GCMs in simulating annual, monsoon, winter, pre-monsoon, and post-monsoon precipitation, maximum and minimum temperature over Pakistan for the period 1961-2005. The quantitative evaluation was conducted using six state-of-the-art spatial metrics; SPAtial EFficiency, Fractions Skill Score, Goodman–Kruskal's lambda, Cramer's V, Mapcurves, and Kling-Gupta efficiency and qualitative evaluation was done using scatter plots. A comprehensive rating metric was used to derive the overall ranks of GCMs based on their ranks pertaining to annual, monsoon, winter, pre-monsoon, and post-monsoon precipitation, maximum and minimum temperature.

Following conclusions were drawn from this study:

1) The low Normalized Root Mean Square Error (NRMSE), and high modified index of agreement (*md*) confirmed the close agreement of monthly Global Precipitation Climatology Center (GPCC) precipitation and Climatic Research Unit (CRU) temperature with the observed precipitation and temperature extracted from 17 stations located in different climate zones in Pakistan. The low NRMSE and high *md* values of GPCC precipitation and CRU temperature can be associated with extensive data quality control measures and the use of a large number of stations for the development of GPCC precipitation and CRU temperature data sets (Schneider et al., 2013;Harris et al., 2014).

2) Ranks of the 36 GCMs derived based on all spatial metrics; SPAtial EFficiency, Fractions Skill Score, Goodman–Kruskal's lambda, Cramer's V, Mapcurves, and Kling-Gupta efficiency for the period 1961-2005 were found mostly similar to each other during a given season (i.e. annual, monsoon, winter, pre-monsoon, and post-monsoon) for a given climate variable (i.e. precipitation, maximum and minimum temperature). However, it was noticed that different GCMs performed significantly differently in simulating different variables (i.e. precipitation, maximum and minimum temperature).

3) EC-EARTH, BCC-CSM1.1 (m) and CSIRO-Mk3-6-0 were identified as the most skilful GCMs while IPSL-CM5B-LR, CMCC-CM, and INMCM4 were identified as the least skilful GCMs in simulating precipitation, maximum and minimum temperature over Pakistan, respectively. The overall ranks of GCMs based on comprehensive rating metric revealed that NorESM1-M, MIROC5, BCC-CSM1-1 and ACCESS1-3 are the most suitable GCMs for simulating all three climate variables (i.e. precipitation, maximum and minimum temperature) over Pakistan.

4) The spatial patterns of precipitation, maximum and minimum temperature of four top-ranked GCMs and their MME mean precipitation, maximum and minimum temperature generated using Simple Mean (SM) and Random Forest (RF) for annual, monsoon, winter, pre-and post-monsoon seasons showed more or less similar spatial patterns to those of GPCC precipitation and CRU maximum and minimum temperature. Moreover, the comparison of MME mean precipitation, maximum and minimum temperature corresponding to annual, monsoon, winter, pre-and post-monsoon seasons generated using Simple Mean (SM) and Random Forest (RF) clearly showed the superiority of Random Forest in replicating the spatial patterns of the GPCC precipitation and CRU maximum and minimum temperature.

*Data availability.*

The model codes and the data are available upon request.

*Author contributions.*

KA, DAS, and SS designed the research and wrote the manuscript. MCD and ESC critically reviewed the paper.

*Competing interests.*

The authors declare that they have no conflict of interest.

*Acknowledgement*

Authors are grateful to the developers of GPCC gridded precipitation and CRU gridded temperature datasets. Authors are also grateful to the IPCC Data Distribution Centre for providing precipitation and temperature datasets simulated by the CMIP5 GCMs.

*Financial support.*

This work was supported by the Professional Development Research University (PDRU) grant no. Q.J130000.21A2.04E10 of Universiti Teknologi Malaysia.

*Review statement.* This paper was edited by Luis Samaniego and reviewed by three anonymous referees.

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
