# Peer review of "Selection of multi-model ensemble of GCMs for the simulation of precipitation, maximum and minimum temperature based on spatial assessment metrics"

_Hydrology and Earth System Sciences, 2018_

## Referee Comment (RC1) · Anonymous Referee #1 · 27 Mar 2019

This article propose an ensemble of GCM model for simulation of precipitation based on spatial assessment metrics. The article presents original research works and outputs. The work is relevant to the interests of the readership of HESS and is well-written. However, there are few issues that need to be addressed. Therefore, authors are encouraged to revise the manuscript accordingly.

1- In Section 3, authors introduce different GCM performance assessment metrics. For almost all parameters except Kling-Gupta, the range of the metric and the meaning of the extreme values are elaborated. To be consistent, it is recommended to revise section 3.1.6 accordingly.

[Figure]

2- In section 3.3, it is highlighted that the RM values for annual, mansoon, and winter precipitations are averaged to derive overall rank for each GCM. Does this approach flatten the effect of extreme cases? was it necessary to average them? How were the individual rankings? Authors need to explain the impact of this approach on their final conclusion.

3- It is needed to give some background knowledge about Random Forest method. Why is it selected? It is needed to give some reasoning for this selection. Also in the results and discussion, more explanation is needed for this method.

4- Section 4.1, it is suggested to present NRMSE formula.

5- To have a better understanding about the site, it would be good to add the location of stations on the map.

6- It is also recommended to highlight the limitations of the study in the discussion part.

7- For figures 4,5, and 7 a performance measure such as r-squared is needed for each scatter plot of observed vs simulated data points.

---

## Short Comment (SC1) · 29 May 2019

This manuscript was written fairly well in terms of its academic originality, and scientific descriptions for introduction, methodology, results and conclusions. Especially, it developed a systematic selection framework for many GCMs from CMIP5 based on various state-of-the-art spatial performance metrics. The selected GCMs showed their capability to mimic the spatial patterns of annual and seasonal precipitation. The most impressive point is to summarize so many relevant articles which were published in recent years. This manuscript is worthwhile to be published in this journal. Nevertheless, the following point should be thought carefully in my opinion. This manuscript focuses

on the simulated data for the very past period (1961-2005). However, the superiority of performances of GCMs for the past period doesn't guarantee the exactness of projection for the future period. In addition, the main objective of GCMs is to support the forecasted future data for 2010-2100. Of course, a part of them can be evaluated using the recent data (2010-2018). If you cannot quantify GCMs' performances for the recent data, you can mention that point in the manuscript. In section 3.3, the four top ranked GCMs were used to generate the most appropriate ensemble of GCMs. Is there any reason why four is used? You can compare your results with those form different numbers of GCMs. This number can affect the results. In section 3.4, you mentioned numerous approaches have been used to calculate the mean time series from an ensemble of better performing GCMs. Thus, it is better to add the reason why two representative methods should be used. What is the improved one? If the method is very critical to the results, you should add the descriptions on simple mean and random forest methods.

Miscellaneous errors When the abbreviation was defined, it should be done at the first appearance. E.g.) p3 L13, root mean square error, p2 L17 multi-model ensemble Check the abbreviation. When any was defined, abbreviation should be used afterwards. E.g. P4 L12, P5 L31 MME, P6 L24 SPAtial EFficiency metric; P6 L8, P10 L7, P16 L24 Rating metric, P11 L5, P15 L19 simple mean and random forest. P7 L12 "lamba"? Check the name of variables in all equations. "N"s in Eq. (5) and (6) are the same? Check the other variables. "(MME)" in the sub-title can be removed. P2 L17 Check Pour et al.(2018b) which is not included in the reference. P2 L22 Check Wang et al.(2017b) which is not included in the reference list. P2 L23 Check Wang et al.(2017a) which is not included in the reference list. P21 L15-16 Salman et al.(2018a) is the same to Salman et al. (2018b), P2 L30 Check Pour et al.(2018b) which is not included in the reference. P3 L8 Are Tebaldi et al. (2005) and Chandler (2013) included in the reference list? P4 L9 "and" should be added at the end of this sentence. P7, P8, P10 Variables "m" and "n" were used in the different equations. Check their consistency. P7 L1 in equation1, is "KGE" correct? SPAEF? P9 L15-16: it should be

moved below Equation 8. P9 L17-18: it should be moved below equation 9 and 10. In the conclusions, abbreviations were defined again. Is it correct in this journal? Check it. P22 L18 Wang et al. (2016) is not cited in the manuscript. Check the reference format.

Grammar P2 L24 "to use" is right? P2 L30, "selection ∼ modelling." is right? P3 L7 "such as" was repeated. P3 L25 "scale" or "scales"? P3 L29 "should able" or "should be able"? P5 L16 the second "20" is not necessary. It was already mentioned at the previous sentence. P5 L30 "are" or "is"? P14 L14 "point" or "points"? P14 L18 "scatter" is right? P14 L16 "skillful"? P14 L17 Check the location of"also".
* * *

---

## Referee Comment (RC2) · Anonymous Referee #2 · 18 Jun 2019

The authors have evaluated precipitation simulation of 20 CMIP5 GCMs for Pakistan, and developed multi-model ensemble mean at annual and seasonal timescales. The topic is relevant to the journal and the findings are interesting. The authors have shown the application of random forests for MME, which is somehow novel. However, the study needs substantial improvement in explaining the methods. The details of some of the methods are missing and should be further explained. Please find more detailed comments in the following:

P refers to page number and L is the line number (please consider using continuous line numbering in future publications).

1. P1, L13: "number metrics" » "number of metrics"

2. P1 L14: "very little attention has been given to spatial performance of GCMs" » Better to rephrase this sentence, since several studies have considered both spatial and temporal characteristics for evaluating GCMs.

3. P2, L4: "land and ocean temperature" » "land and ocean surface air temperature"

4. P2, L26: "better GCMs are assigned higher weightages" » "higher weights are assigned to better GCMs"

5. P2, L31: "is climate change modelling" » "in climate change modelling"

6. P3, L7: "such as such as" » please remove the redundant "such as".

7. P5, L25: "five ... measures" » six measures are introduced here. Please revise the number.

8. P6, L4-5: did you apply these measures on each grid? Or are they applied on temporally averaged data? For instance, how is KGE calculated? Please explain.

9. P5, L8-9: "comprehensive rating metric" » what does this indicate? How were the ranks of GCMs (from different measures) combined? Do you mean like averaging the ranks? If so, "comprehensive" is misleading and it is better to be revised.

10. P6, L28: Eq (1) seems to be the equation for KGE. Please make sure to provide the equation for SPAEF here.

11. Section 3.1.1: I am still not sure how the measure is calculated? Is it applied to each grid ($2°\text{x}2°$), and then maybe the spatial mean value of KGE is considered? Or, did you take the long-term average of precipitation and then calculate the KGE for a few grids?

12. Section 3.1.5 is not clearly explained. A, B, and C need more explanation. What do you mean by "total area of historical and GCM simulated maps"? Is this the coverage

area? If so, then both GCM and obs have the same number of grids and the areas should be identical in all cases. In addition, what does "the degree of intersection (for C)" refer to? How can one quantify such thing? Is there a function for calculating it? These need to be clearly explained.

13. P11, l2: "can be reduce" » "can be reduced"

14. Section 3.4: There is no explanation about the details of the random forest method. How many trees were included? What are the inputs to the model (time series of 4 selected GCMs?)? Is the model applied separately for each grid, or did you employ a consistent model for the entire study domain? How long is the training and testing periods? How did you evaluate the performance of the random forest outputs?

15. Figure 4 caption: Does the figure show long-term average values for different grids? Or, does it show spatial mean precipitation in various years. Please clarify it in the caption and the text, and mention the period for it as well.

---

## Referee Comment (RC3) · Anonymous Referee #3 · 9 Jul 2019

In this manuscript, authors evaluated precipitation data from 20 CMIP5 GCMs and selected four better-performing CMIP5 GCMs based on their spatial performance against observed precipitation (GPCC) during the historical period (1961-2005). To evaluate the skill of model precipitation (CMIP5 GCMs) against observed precipitation (GPCC), they used six spatial metrics (SPAEF, Goodman-Kruskal's lambda, Fractions Skill Score, Cramer's V, Mapcurves, and Kling-Gupta efficiency). Finally, they generated multi-model ensemble mean (MME) of precipitation of four selected GCMs using Random forest regression and simple mean method. The manuscript is written fairly well, and the idea of spatial assessment of CMIP5 GCMs for multi-model ensemble mean is appreciated. However, the execution of manuscript seems sloppy and hasty.

[Figure]

There are numerous methodological, data, explanation, reporting, and citation issues in the manuscript. Thus I recommend major revisions be required before publication.

Major issues:

1. Error and unexplained parameters in the formula of matrices: I have many doubts about spatial assessment methods. Authors need to explain all six methods clearly and correctly. a) In Goodman-Kruskal's lambda, how many classes you have taken in the contingency matrix? Please mention the number of classes and explain- Are these classes sufficient to explain spatial variability of rainfall or measure the matrix accurately? Did you consider only one annual map to estimate the lambda value for each model? If yes, then there may be many years those have low or high bias but not captured in the annual mean map. You need to estimate lambda value for each year or seasonal map. What is the maxj (or maxj)? What is the value of m and n? b) In the fraction skill score, there should Nx*Ny in the palace of N. Roberts and Lean, (2008) used Nx*Ny. It will affect the final results. Please explain it. c) In Cramer's V, you have taken the wrong formula. There should be $N*(min(m-1,n-1))$, but you have taken $N*(min(m,n)-1$. It will also affect your final selection. d) In Mapcurves method, did you classify your map in the different range of rain? If yes, how many classes you have taken? Did you calculate Y value for each month/ season/year? It should be calculated for each year (1961-2005) between model and GPCC data in the case of annual values. e) In Kling-Gupta efficiency, please check Demirel et al., 2018 paper. They have taken different formulas for beta and gamma. f) Why did you choose these six methods? What are the limitations of each method? Please explain.

2. Error in rating metrics formula: (P10, L10) In this formula, rank varies from 1 to 6 (n=6) but it should be 1 to 20 (model=20) for each matrix. Please explain this.

3. Pre-monsoon and Post-monsoon seasons: Why did you not consider the pre and post monsoon season for the analysis and during the overall rank. These seasons will affect significantly in the overall ranking. I recommend to estimate rank month-wise.

[Figure]

That will improve the results significantly and should not provide the same weight to each month. Here, you provided the same weight to annual, monsoon, and winter rank (during overall rank). Why?

4. Inconsistency in spatial resolution: You should consider the same spatial resolution to compare the maps or data sample. In the manuscript, observation data (GPCC) are available at $0.5°$ resolution and model data are prepared at $2°$ resolution. Model data should be regridded at $2°$.

5. Random Forest Method: Please explain the method and weight value.

6. Increase the number of CMIP5 models in the study: Authors used only 20 models for the current study and said all four RCP data available for 20 models. However, there is no use of RCP data in the analysis. Hence, they can get historical data for more than 35 CMIP5 GCMs. That will increase the scope and use of this study. I recommend they should use the maximum number of models.

7. Selection of better performing models should be based on at least precipitation and temperature: In the manuscript, authors used only precipitation variable to select better performing models, but there are many models under CMIP5 those have low projection skill in temperature data and high skill in precipitation. Hence, there is a possibility of the poor skill of temperature projection in the selected GCMs. Moreover, most of the studies in the hydrology and earth science commonly use precipitation and temperature variables. Therefore, they should include the temperature variable in the analysis and select the models based on the high skill in both (Precipitation and temperature) variables.

Other issues: (P- no of page; L- no of line) 1. P2, L1 – please provide citation after several studies (related to the heatwaves, cold snaps etc.). Duffy et al. (2015) is about drought and wet spells. 2. P2, L7: please provide the correct citation. Hegerl et al., 2018 is not about the affecting hydrological cycle (that include ET, runoff, soil moisture, and precip) 3. P2, L9: should be Akhter et al., 2017\ 4. P2, L10: Wright et al., 2015 is

about RCMs. Please provide a correct reference. 5. P2, L13: cite CMIP5 GCMs 6. P2, L14: Cited paper is not about the cmip5 and cmip3 comparison. 7. P2, L14: more than 50 GCMS are available. Please check other papers. 8. P2, L16: Ekstrom et al., 2016 is not about size and restriction on the size of the subset of GCMs. 9. P2, L16: Salam et al., 2018a and 2018b is same 10. P2, L17: should be 2018 11. P2, L16: cite some paper about the uncertainties in GCMs and why do we need to do ensemble mean. Please add some line about this. 12. P2, L19: "prediction" ("projection") 13. P2, L22: Wang et al., 2017 14. P2, L24: Wang et al., 2017 15. P2, L25: Fu et al., 2018 and Dong et al., 2018 are not about the comparison between MEE and individual. They are based on temperature projection. 16. P2, L31: 2018 17. P3, L15: Gleckler et al., 2008a and 2018b are same. 18. P4, L1: provide citation after several studies. 19. P4, L7: you used six methods. Please correct this number throughout the paper. 20. P4, L11: please mention the calendar months. 21. Figure 1: should include a climate zone map also. 22. P4, L25-29: this data conflict with the fig 3a. 23. P5, L7: please provide the website link (GPCC data). 24. P5, L11: high correlation? Please provide the number. 25. P5, L14-20: Please mention the ensemble member that you have used in the CMIP5 GCMs. 26. P5, L15: provide a website link. 27. P6, L24: Please check the citation. In the introduction, you mentioned Demirel et al., (2018). 28. P7, L11: Lambda (heading) 29. P11, L12- 25: You did not mention about the time series. Is it annual rainfall or seasonal or monthly time series? Did you check NRMSE and md between the annual time series? 30. No need for figure 2. You can remove the figure 2 and include the rank in table 3 in brackets.

---

## Author Comment (AC1) · 26 Aug 2019

General Comments This manuscript was written fairly well in terms of its academic originality, and scientific descriptions for introduction, methodology, results and conclusions. Especially, it developed a systematic selection framework for many GCMs from CMIP5 based on various state-of-the-art spatial performance metrics. The selected GCMs showed their capability to mimic the spatial patterns of annual and seasonal precipitation. The most impressive point is to summarize so many relevant articles which were published in recent years. This manuscript is worthwhile to be published in this journal. Nevertheless, the following point should be thought carefully in my opinion.

[Figure]

This manuscript focuses on the simulated data for the very past period (1961-2005). However, the superiority of performances of GCMs for the past period doesn't guarantee the exactness of projection for the future period. In addition, the main objective of GCMs is to support the forecasted future data for 2010-2100. Of course, a part of them can be evaluated using the recent data (2010-2018). If you cannot quantify GCMs' performances for the recent data, you can mention that point in the manuscript. Reply Thank you for your constructive comments on our manuscript. Your suggestions helped us to improve the quality of our manuscript. In response to your above comment we have added following paragraph as a limitation in the discussion of the manuscript. "In this study performance of GCMs was assessed based on their ability to simulate past observed P, Tmax and Tmin and hence the best performing GCMs were identified and used for the development of MMEs. However, it is found that past and future climate may have a weak association hence it is not necessary that if a GCM performs well in the past will give reliable results in future (Knutti et al., 2010). In other words, the best GCMs selected for the MMEs considering their ability to simulate past climate may not be the best in the future under changing climate (Ruane and McDermid, 2017;Ahmed et al., 2019b). This is due to the large uncertainties associated with GHG emission scenarios and GCMs. As a solution to this limitation, Salman et al. (2018) selected an ensemble of GCMs based on past performance as well as the degree of agreement between their future projections. The study detailed in the present manuscript can be repeated in future to select GCMs considering their past performance and the degree of agreement in their future projections. . Comment 1 In section 3.3, the four top ranked GCMs were used to generate the most appropriate ensemble of GCMs. Is there any reason why four is used? You can compare your results with those form different numbers of GCMs. This number can affect the results.

Reply Thank you very much for your very interesting comment. As seen in the literature the number of GCMs selected for the MME is an arbitrary choice. The choice for the selection of four top GCMs in this study is also arbitrary. We have added the following text to section "3.4 Identification of Ensemble Members" of the manuscript as below: "3.4

Identification of Ensemble Members The uncertainties in climate projections arise from GCM structure, assumptions and approximations, initial conditions, and parameterization can be reduced by identifying an ensemble of better performing GCMs (Kim et al., 2015). Lutz et al. (2016) reported that one or a small ensemble of GCMs is suitable for climate change impact assessment. A number of studies (Weigel et al., 2010;Miao et al., 2012) have suggested that one GCM is not enough to assess the uncertainties associated with the future climate. Therefore, identification of an ensemble of GCMs is a necessity in climate change impact assessment. In the present study, four top ranked GCMs were considered for the development of MMEs for P, Tmax and Tmin. The review of the literature revealed that there is no well-defined guideline on the selection of the optimum number of GCMs for the MME and most of the studies considered the first three to ten GCMs ranked according to the descending order of their performance for the MME. For instance, in the study by Xuan et al. (2017) over Zhejiang, China, ten top-ranked GCMs for an MME for precipitation were used. In another study over China, Jiang et al. (2015) developed MMEs for daily temperature extremes using the five top-ranked GCMs. In a study over Pakistan, Khan et al. (2018) considered six common GCMs that appeared in the lists of ten top-ranked GCMs for daily temperature and precipitation. Ahmadalipour et al. (2015) used the four top-ranked GCMs for simulating daily precipitation and temperature over the Columbia River Basin in the Pacific Northwest USA. In the study by Hussain et al. (2018) the three top-ranked GCMs for the development of an MME for precipitation over Bornean tropical rainforests in Malaysia were used. In the present study, the ensemble of GCMs was identified in two steps: (1) RM values of GCMs for annual, monsoon, winter, pre-monsoon and post-monsoon P, Tmax and Tmin were individually used to derive an overall rank for each GCM, and (2) four top ranked GCMs based on RM values for all variables were considered for the ensemble. The selection of an appropriate set of GCMs considering their skills in different seasons enables the selection of an ensemble which can better simulate the observations in different seasons."

Following lines are added as the limitation of the study in discussion section. "In the

present study, the MME of P, Tmax and Tmin were developed by considering top four ranked GCMs. In the past, MMEs were developed considering 3 to 10 top ranked GCMs. However, none of the study showed the performance of MME by varying the number of GCMs in MME. It can be remarked that the performance of an MME is probably sensitive to the choice of the GCMs. Hence, in future, a study should be conducted to investigate the impact of the number of GCMs used for the development of the MME on its performance."

Comment 2 In section 3.4, you mentioned numerous approaches have been used to calculate the mean time series from an ensemble of better performing GCMs. Thus, it is better to add the reason why two representative methods should be used. What is the improved one? If the method is very critical to the results, you should add the descriptions on simple mean and random forest methods.

Reply Thanks for your comment. We have already addressed the above comment partly in the introduction and section 3.4 of the manuscript as given below. In addition to that, we have now added details on simple mean and random forest to sections 3.4 of the revised manuscript. In introduction section: "The methods used for the generation of MME are broadly divided into two groups; (1) simple composite method (SCM) and (2) weighted ensemble method (WEM) (Wang et al., 2018). In SCM all ensemble members are equally weighted while in the WEM, ensemble members are weighted according to their performance in simulating the past climate (Wang et al., 2018;Oh and Suh, 2017;Giorgi and Mearns, 2002). The SCM is relatively simple to apply and found to perform better than individual GCMs (Weigel et al., 2010;Acharya et al., 2013;Wang et al., 2018). However, WEM is preferred as it has the capability to remove the systematic biases and improve the prediction capability since higher weights are assigned to better GCMs (Krishnamurti et al., 1999;Krishnamurti et al., 2000). Salman et al. (2018) reported that prediction capability of a MME improves if it is based on WEM method. Thober and Samaniego (2014) also showed that sub-ensembles generated using WEM has the better capability to capture the historical characteristics of precipitation and temperature extremes. The performances of MMEs depend on the performance of ensemble members in simulating historical climate (Pour et al., 2018). Therefore, selection of a sub-ensemble is a major challenge in climate change modelling." We revised section 3.4 as below: "The uncertainties in climate projections arise from GCM structure, assumptions and approximations, initial conditions, and parameterization can be reduced by identifying an ensemble of better performing GCMs (Kim et al., 2015). Lutz et al. (2016) reported that one or a small ensemble of GCMs is suitable for climate change impact assessment. A number of studies (Weigel et al., 2010;Miao et al., 2012) have suggested that one GCM is not enough to assess the uncertainties associated with the future climate. Therefore, identification of an ensemble of GCMs is a necessity in climate change impact assessment. In the present study, four top ranked GCMs were considered for the development of MMEs for P, Tmax and Tmin. The review of the literature revealed that there is no well-defined guideline on the selection of the optimum number of GCMs for the MME and most of the studies considered the first three to ten GCMs ranked according to the descending order of their performance for the MME. For instance, in the study by Xuan et al. (2017) over Zhejiang, China, ten top-ranked GCMs for an MME for precipitation were used. In another study over China, Jiang et al. (2015) developed MMEs for daily temperature extremes using the five top-ranked GCMs. In a study over Pakistan, Khan et al. (2018) considered six common GCMs that appeared in the lists of ten top-ranked GCMs for daily temperature and precipitation. Ahmadalipour et al. (2015) used the four top-ranked GCMs for simulating daily precipitation and temperature over the Columbia River Basin in the Pacific Northwest USA. In the study by Hussain et al. (2018) the three top-ranked GCMs for the development of an MME for precipitation over Bornean tropical rainforests in Malaysia were used. In the present study, the ensemble of GCMs was identified in two steps: (1) RM values of GCMs for annual, monsoon, winter, pre-monsoon and post-monsoon P, Tmax and Tmin were individually used to derive an overall rank for each GCM, and (2) four top ranked GCMs based on RM values for all variables were considered for the ensemble. The selection of an appropriate set of GCMs considering their skills in

different seasons enables the selection of an ensemble which can better simulate the observations in different seasons.

Details of both Simple Mean and Random Forest used in developing MMEs are given below.

3.5.1 Simple Mean (SM) Simple Mean (SM)-based MMEs were developed by simply averaging the individual P, Tmax and Tmin simulations of the four top-ranked GCMs using Eq.16.

SM=1/n $\sum\_(i=1)$ $nGCM\_i (16)$

In Eq. 16, n refers to the number of GCMs considered for the development of MMEs which is four in the present study and GCMi refers to the P, Tmax and Tmin simulation of the ith GCM.

3.5.2 Random Forest (RF) Random Forest (RF) algorithm (Breiman, 2001) was used in the calculation of the mean time series of P, Tmax and Tmin corresponding to an MME of four top ranked GCMs. RF is a relatively new machine learning algorithm widely used in modelling non-linear relationships between predictors and predictands (Ahmed et al., 2019a). RF algorithm is found to perform well with spatial data sets and less prone to over-fitting (Folberth et al., 2019). Most importantly Folberth et al. (2019) reported that RF is less sensitive to multivariate correlation. RF is an ensemble technique where regression is done using multiple decision trees. RF algorithm uses the following steps in regression. A bootstrap resampling method is used to select sample sets from training data. Classification And Regression Tree (CART) technique is used to develop unpruned trees using the bootstrap sample. A large number of trees are developed with the samples selected repetitively from training data so that all training data have equal probability of selection. A regression model is fitted for all the trees and the performance of each tree is assessed. Ensemble prediction is estimated by averaging the predictions of all trees which is considered as the final prediction. Wang et al. (2017) and He et al. (2016) reported that the performance of RF varies

with the number of trees (ntree) and the number of variables randomly sampled at each split in developing the trees (mtry). It was observed that RF performance increases with the increase in ntree. However, in the present study the performance was not found to increase significantly in term of root mean square error when the ntree was greater than 500. Therefore, ntree was set to 500 while the mtry was set to p/3 where p is the number of variables (i.e. GCMs) used for developing RF-based MME. The MME prediction can be improved by assigning larger weight to the GCMs which show better performance (Sa'adi et al., 2017). RF regression models developed using historical P, Tmax and Tmin simulations of GCMs as independent variable and historical observed P, Tmax and Tmin as dependent variable provide weights to the GCMs according to their ability to simulate historical observed P, Tmax and Tmin. The "Random Forest" package written in R programming language was employed in this study for developing RF-based MMEs. RF-based MMEs were calibrated with the first 70% of the data and validated with the rest of the data.

Comment 3 When the abbreviation was defined, it should be done at the first appearance. E.g.) p3 L13, root mean square error, p2 L17 multi-model ensemble. Check the abbreviation. When any was defined, abbreviation should be used afterwards. E.g. P4 L12, P5 L31 MME, P6 L24 SPAtial EFficiency metric; P6 L8, P10 L7, P16 L24 Rating metric, P11 L5, P15 L19 simple mean and random forest.

Reply Thanks. Corrected as suggested.

Comment 4 P7 L12 "lamba"? Check the name of variables in all equations. "N"s in Eq. (5) and (6) are the same? Check the other variables. "(MME)" in the sub-title can be removed.

Reply Thanks, "lamba" was corrected as "lambda". The "N" in Eq. 5 and 6 are different and defined differently. "MME" in the subtitle was removed.

Comment 5 P2 L17 Check Pour et al.(2018b) which is not included in the reference.

Reply Thanks, we have updated reference list. It is corrected now.

Comment 6 P2 L22 Check Wang et al.(2017b) which is not included in the reference list. Reply Thanks, we have updated reference list. It is corrected now.

Comment 7 P2 L23 Check Wang et al.(2017a) which is not included in the reference list. Reply Thanks, we have updated reference list. It is corrected now.

Comment 8 P21 L15-16 Salman et al.(2018a) is the same to Salman et al. (2018b), Reply Thanks, we have updated reference list. It is corrected now.

Comment 9 P2 L30 Check Pour et al.(2018b) which is not included in the reference. Reply Thanks, we have updated reference list. It is corrected now.

Comment 10 P3 L8 Are Tebaldi et al. (2005) and Chandler (2013) included in the reference list? Reply Yes, they are also included in revised reference list.

Comment 11 P4 L9 "and" should be added at the end of this sentence. Reply Thanks, we have added "and" at the end of the sentence.

Comment 12 P7, P8, P10 Variables "m" and "n" were used in the different equations. Check their consistency. Reply Thanks for the comment, we have re-checked, "m" and "n" were used in the different equations and they are defined in accordingly.

Comment 13 P7 L1 in equation1, is "KGE" correct? SPAEF? Reply Thanks, it was corrected.

Comment 14 P9 L15-16: it should be moved below Equation 8. Reply Thanks for this comment. P9 L15-16 are moved below equation 8.

Comment 15 P9 L17-18: it should be moved below equation 9 and 10. In the conclusions, abbreviations were defined again. Is it correct in this journal? Check it. Reply Thanks for this comment. P9 L17-18 are moved below equation 9 and 10. The journal does not have any restriction on the use of long-terms of abbreviations in the conclusions. We have defined some of the abbreviations in the conclusion as it may assist

a reader who wants to have a glance at the conclusions and understand the main outcomes of the work.

Comment 16 P22 L18 Wang et al. (2016) is not cited in the manuscript. Check the reference format. Reply Thanks, we have updated reference list. It is corrected now.

Comment 17 P2 L24 "to use" is right? Reply We have changed the phrase "to use" to " to apply"

Comment 18 P2 L30, "selection âĹij modelling." is right? Reply The word "selection" is more appropriate here as we are referring to the selection of GCMs.

Comment 19 P3 L7 "such as" was repeated. Reply Thanks. Corrected as suggested.

Comment 20 P3 L25 "scale" or "scales"? Reply Thanks, we changed scale to scales.

Comment 21 P3 L29 "should able" or "should be able"? Reply Thanks, "should able" was replaced with "should be able".

Comment 22 P5 L16 the second "20" is not necessary. It was already mentioned at the previous sentence. Reply Thanks, the second "20" was removed from the next sentence.

Comment 23 P5 L30 "are" or "is"? Reply Thanks, We have changed "are" to "is".

Comment 24 P14 L14 "point" or "points"? Reply Thanks, we changed point to points.

Comment 25 P14 L18 "scatter" is right? Reply Yes, it is correct.

Comment 26 P14 L16 "skillful"? Reply Thanks, skilfull is changed to skillful.

Comment 27 P14 L17 Check the location of"also". Reply Thanks, we revised sentence as shown below. "Over and underestimation of precipitation can also be seen in the scatter. . .."

References

[revised manuscript text omitted]

---

## Author Comment (AC2) · 26 Aug 2019

Comment This article proposes an ensemble of GCM model for simulation of precipitation based on spatial assessment metrics. The article presents original research works and outputs. The work is relevant to the interests of the readership of HESS and is well-written. However, there are few issues that need to be addressed. Therefore, authors are encouraged to revise the manuscript accordingly.

Reply Thanks for your highly constructive comments on our manuscript. The manuscript has now been revised according to the comments. The details of the revisions made are given under each comment. Revisions are marked in Red.

[Figure]

Comment 1 In Section 3, authors introduce different GCM performance assessment metrics. For almost all parameters except Kling-Gupta, the range of the metric and the meaning of the extreme values are elaborated. To be consistent, it is recommended to revise section 3.1.6 accordingly.

Reply Thanks for your suggestion. We have now defined the range of KGE as below. "In the present study, KGE was calculated between historical observed data and GCM simulated data using Eq. (12). KGE values can range between –infinity to 1, where values closer to 1 are preferred."

Comment 2 In section 3.3, it is highlighted that the RM values for annual, monsoon, and winter precipitations are averaged to derive overall rank for each GCM. Does this approach flatten the effect of extreme cases? Was it necessary to average them? How were the individual rankings? Authors need to explain the impact of this approach on their final conclusion.

Reply Thank you very much for this interesting comment. In the original manuscript, in order to derive an overall rank for each GCM, RM values corresponding to annual, monsoon and winter precipitation were first averaged and then based on the average of RM values an overall rank was assigned to each GCM. This procedure helped in assigning one single rank to each GCM while taking into account precipitation for annual, monsoon, and winter seasons all together. Following your above comment, in the revised manuscript, we ranked each GCM for each season (i.e. annual, monsoon, winter, pre-monsoon, and post-monsoon precipitation) to derive ranks for each variable (precipitation, maximum and minimum temperature) separately by applying comprehensive rating metric. Later, comprehensive rating metric was again applied on precipitation, maximum and minimum temperature ranks to derive an overall rank of GCMs for the whole study area. This procedure helps us to avoid averaging. The obtained results are discussed in section 4.3 as below.

4.3 Overall Ranks of GCMs for Precipitation, Maximum Temperature and Minimum

Temperature The application of various evaluation metrics has yielded different ranks for the same GCM (Ahmadalipour et al., 2017;Raju et al., 2017). The ranks attained by GCMs corresponding to different metrics and seasons (annual, monsoon, winter, pre-monsoon and post-monsoon) were used to calculate the RM values for each GCM. The ranks of GCMs for P, Tmax and Tmin are presented in Table 4 along with the RM values. As seen in Table 4, EC-EARTH, BCC-CSM1.1 (m) and CSIRO-Mk3-6-0 were the most skillful GCMs in reproducing the spatial characteristics of P, Tmax and Tmin respectively. On the other hand, IPSL-CM5B-LR, CMCC-CM, and INMCM4 displayed the least skill in reproducing the spatial characteristics of P, Tmax and Tmin respectively.

Table 4 in the supplement file

The better performance of EC-EARTH, BCC-CSM1.1 (m) and CSIRO-Mk3-6-0 in simulating P, Tmax and Tmin over Indo-Pak sub-continent has also been reported in several past studies. Latif et al. (2018) reported the relatively better performance of EC-EARTH, and BCC-CSM1.1 (m) out of 36 CMIP5 GCMs in simulating precipitation over Indo-Pakistan sub-continent based on spatial correlation. Rehman et al. (2018) conducted a study to assess the performance of CMIP5 GCMs in simulating the mean precipitation and temperature over south Asia. The study reported the better performance of EC-EARTH in simulating precipitation and CSIRO-Mk3-6-0 in simulating temperature. Khan et al. (2018) assessed the performance of 31 CMIP5 GCMs in simulating the mean precipitation and temperature over Pakistan using multiple daily gridded datasets and identified EC-EARTH as the best GCM for simulating precipitation and CSIRO-Mk3-6-0 for simulating temperature. Better performance of CSIRO-Mk3-6-0 in simulating maximum and minimum temperature is also reported in the study by (Ahmed et al., 2019b).

Regarding the issue of deriving rank based on average of RM values (given in the original manuscript) and overall ranks based on individual ranks; a comparison was made. The comparison of overall ranks obtained with the above two approaches is

shown in Table below (not included in the manuscript). As seen in Table below, it was understood that the differences between the overall ranks derived based on; average of RM values and overall ranks is mostly quite small. Therefore, it can be stated that either the overall ranks can be derived based on average of RM values or individual ranks. However, derivation of overall ranks based on ranks of individual variable is relatively simple and hence recommended.

The 2nd Table in the Supplement file.

Comment 3 It is needed to give some background knowledge about Random Forest method. Why is it selected? It is needed to give some reasoning for this selection. Also in the results and discussion, more explanation is needed for this method. Reply Thank you very much for this comment. We have now added a new sub-section covering the information related to Random Forest algorithm as show below. . 3.5.2 Random Forest (RF) Random Forest (RF) algorithm (Breiman, 2001) was used in the calculation of the mean time series of P, Tmax and Tmin corresponding to an MME of four top ranked GCMs. RF is a relatively new machine learning algorithm widely used in modelling non-linear relationships between predictors and predictands (Ahmed et al., 2019a). RF algorithm is found to perform well with spatial data sets and less prone to over-fitting (Folberth et al., 2019). Most importantly Folberth et al. (2019) reported that RF is less sensitive to multivariate correlation. RF is an ensemble technique where regression is done using multiple decision trees. RF algorithm uses the following steps in regression. A bootstrap resampling method is used to select sample sets from training data. Classification And Regression Tree (CART) technique is used to develop unpruned trees using the bootstrap sample. A large number of trees are developed with the samples selected repetitively from training data so that all training data have equal probability of selection. A regression model is fitted for all the trees and the performance of each tree is assessed. Ensemble prediction is estimated by averaging the predictions of all trees which is considered as the final prediction. Wang et al. (2017) and He et al. (2016) reported that the performance of RF varies with the number of

trees (ntree) and the number of variables randomly sampled at each split in developing the trees (mtry). It was observed that RF performance increases with the increase in ntree. However, in the present study the performance was not found to increase significantly in term of root mean square error when the ntree was greater than 500. Therefore, ntree was set to 500 while the mtry was set to p/3 where p is the number of variables (i.e. GCMs) used for developing RF-based MME. The MME prediction can be improved by assigning larger weight to the GCMs which show better performance (Sa'adi et al., 2017). RF regression models developed using historical P, Tmax and Tmin simulations of GCMs as independent variable and historical observed P, Tmax and Tmin as dependent variable provide weights to the GCMs according to their ability to simulate historical observed P, Tmax and Tmin. The "Random Forest" package written in R programming language was employed in this study for developing RF-based MMEs. RF-based MMEs were calibrated with the first 70% of the data and validated with the rest of the data.

Comment 4 Section 4.1, it is suggested to present NRMSE formula. Reply Thank you very much for your suggestion. We have now added a sub-section entitled "Accuracy Assessment of Gridded Precipitation Data" under the method section 3.1 and provided details on NRMSE and md. 3.1 Accuracy Assessment of Gridded Precipitation and Temperature Data The accuracy of gridded GPCC precipitation data and CRU temperature data was assessed by comparing them with observed station data using NRMSE and md. NRMSE is a non-dimensional form of Root Mean Square Error (RMSE) which is derived by normalizing RMSE by variance of observations. NRMSE is more reliable than RMSE in comparing model performance when the model outputs are in different units or the same unit but with different orders of magnitude (Willmott, 1982). NRMSE can have any positive value, however values closer to 0 are preferred as they denote smaller errors (Chen and Liu, 2012). In this study, NRMSE was calculated Eq. 1.

NRMSE=( )/(x_max-x_min ) (1) Where xsim,i and xobs,i refer to the ith value in the gridded and observed time series of the climate variable (i.e. precipitation or temperature)

[Figure]

respectively, and N is the number of data points in each time series.

The 'md' shown in Eq. 2 is widely used to estimate the agreement between observed and gridded data of climate variables (Noor et al., 2019;Ahmed et al., 2019a). It varies between 0 (no agreement) and 1 (perfect agreement) (Willmott, 1981).

(2)

Where xsim,i and xobs,i are the ith data point in the gridded data and observed data series of a climate variable.

Comment 5 To have a better understanding about the site, it would be good to add the location of stations on the map. Reply Thank you very much for your suggestion. We have revised Figure 1 and included the locations of the stations. Also, we have provided the names of stations in Table 2.

Figure 1 in the Supplement file.

Table 2 in the Supplement file.

Comment 6 It is also recommended to highlight the limitations of the study in the discussion part Reply Thanks for your suggestion. We have now added the following paragraph to the manuscript to highlight the limitations of this study and recommendation for future work. "In this study performance of GCMs was assessed based on their ability to simulate past observed P, Tmax and Tmin and hence the best performing GCMs were identified and used for the development of MMEs. However, it is found that past and future climate may have a weak association hence it is not necessary that if a GCM performs well in the past will give reliable results in future (Knutti et al., 2010). In other words, the best GCMs selected for the MMEs considering their ability to simulate past climate may not be the best in the future under changing climate (Ruane and McDermid, 2017;Ahmed et al., 2019b). This is due to the large uncertainties associated with GHG emission scenarios and GCMs. As a solution to this limitation, Salman et al. (2018) selected an ensemble of GCMs based on past performance as well as the degree of agreement between their future projections. The study detailed in the present manuscript can be repeated in future to select GCMs considering their past performance and the degree of agreement in their future projections. In the present study, the MME of P, Tmax and Tmin were developed by considering top four ranked GCMs. In the past, MMEs were developed considering 3 to 10 top ranked GCMs. However, none of the study showed the performance of MME by varying the number of GCMs in MME. The performance of an MME can be sensitive to the choice of the number of GCMs. Hence, in future, a study should be conducted to investigate the impact of the number of GCMs used for the development of the MME. Only RF algorithm was used in this study for the development of MMEs. Other machine learning algorithms (e.g. Artificial Neural Networks, Support Vector Machine, Relevance Vector Machine, K-nearest neigbour, Extreme Learning method) can also be used for the development of MMEs. A comparison of the performance of MMEs developed with different machine learning algorithms can assist in identification of pros and cons of different algorithms in relation to development of MMEs. In the present study, GCM ranking and MME development was conducted only considering P, Tmax and Tmin pertaining to annual, monsoon, winter, pre-monsoon and post-monsoon seasons. However, several studies reported that the ranking of GCMs based on different climate variables may assist in the identification of a more dependable set of GCMs for ensemble generation (Johnson and Sharma, 2012;Xuan et al., 2017). In future, the ranking of GCMs can be conducted considering several climate variables (e.g. precipitation, mean temperature, maximum temperature, minimum temperature, wind speed, evapotranspiration and solar radiation)."

Comment 7 For figures 4, 5, and 7 a performance measure such as r-squared is needed for each scatter plot of observed vs simulated data points. Reply Thanks for your suggestion. We have indicated the performance of MMEs in terms of the modified index of agreement (md) in all plots in Figures 4, 5 and 7 as shown below.

Figure 4 in the Supplement file.

Figure 5 in the Supplement file.

Figure 7 in the Supplement file.

References

[revised manuscript text omitted]

Please also note the supplement to this comment:
https://www.hydrol-earth-syst-sci-discuss.net/hess-2018-585/hess-2018-585-AC2-supplement.pdf
* * *
[Figure]

**Supplement:**

**Comment**

This article proposes an ensemble of GCM model for simulation of precipitation based on spatial assessment metrics. The article presents original research works and outputs. The work is relevant to the interests of the readership of HESS and is well-written. However, there are few issues that need to be addressed. Therefore, authors are encouraged to revise the manuscript accordingly.

**Reply**

Thanks for your highly constructive comments on our manuscript. The manuscript has now been revised according to the comments. The details of the revisions made are given under each comment. Revisions are marked in Red.

**Comment 1**

In Section 3, authors introduce different GCM performance assessment metrics. For almost all parameters except Kling-Gupta, the range of the metric and the meaning of the extreme values are elaborated. To be consistent, it is recommended to revise section 3.1.6 accordingly.

**Reply**

Thanks for your suggestion. We have now defined the range of KGE as below.

"In the present study, KGE was calculated between historical observed data and GCM simulated data using Eq. (12). KGE values can range between –infinity to 1, where values closer to 1 are preferred."

**Comment 2**

In section 3.3, it is highlighted that the RM values for annual, monsoon, and winter precipitations are averaged to derive overall rank for each GCM. Does this approach flatten the effect of extreme cases? Was it necessary to average them? How were the individual rankings? Authors need to explain the impact of this approach on their final conclusion.

**Reply**

Thank you very much for this interesting comment. In the original manuscript, in order to derive an overall rank for each GCM, RM values corresponding to annual, monsoon and winter precipitation were first averaged and then based on the average of RM values an overall rank was assigned to each GCM. This procedure helped in assigning

one single rank to each GCM while taking into account precipitation for annual, monsoon, and winter seasons all together.

Following your above comment, in the revised manuscript, we ranked each GCM for each season (i.e. annual, monsoon, winter, pre-monsoon, and post-monsoon precipitation) to derive ranks for each variable (precipitation, maximum and minimum temperature) separately by applying comprehensive rating metric. Later, comprehensive rating metric was again applied on precipitation, maximum and minimum temperature ranks to derive an overall rank of GCMs for the whole study area. This procedure helps us to avoid averaging. The obtained results are discussed in section 4.3 as below.

[revised manuscript text omitted]

Regarding the issue of deriving rank based on average of RM values (given in the original manuscript) and overall ranks based on individual ranks; a comparison was made. The comparison of overall ranks obtained with the above two approaches is shown in Table below (not included in the manuscript). As seen in Table below, it was understood that the differences between the overall ranks derived based on; average of RM values and overall ranks is mostly quite small. Therefore, it can be stated that either the overall ranks can be derived based on average of RM values or individual ranks. However, derivation of overall ranks based on ranks of individual variable is relatively simple and hence recommended.

| GCM | $P$ | $T_{max}$ | $T_{min}$ | Avg RM value | Rank based on average RM value | Rank based on overall RM values | Difference |
|---|---|---|---|---|---|---|---|
| ACCESS1-0 | 0.555 | 0.577 | 0.481 | 0.537 | 17 | 13 | 4 |
| ACCESS1-3 | 0.555 | 0.520 | 0.656 | 0.577 | 9 | 4 | 5 |
| BCC-CSM1-1 | 0.506 | 0.534 | 0.413 | 0.484 | 6 | 3 | 3 |
| BCC-CSM1.1(m) | 0.394 | 0.702 | 0.684 | 0.594 | 22 | 23 | -1 |
| BNU-ESM | 0.337 | 0.538 | 0.569 | 0.481 | 23 | 24 | -1 |
| CanESM2 | 0.406 | 0.442 | 0.577 | 0.475 | 25 | 26 | -1 |
| CCSM4 | 0.689 | 0.466 | 0.631 | 0.595 | 5 | 7 | -2 |
| CESM1-BGC | 0.467 | 0.459 | 0.628 | 0.518 | 18 | 19 | -1 |
| CESM1-CAM5 | 0.654 | 0.371 | 0.595 | 0.540 | 14 | 18 | -4 |
| CESM1-WACCM | 0.482 | 0.522 | 0.356 | 0.454 | 27 | 29 | -2 |
| CMCC-CM | 0.353 | 0.249 | 0.428 | 0.343 | 36 | 36 | 0 |
| CMCC-CMS | 0.381 | 0.616 | 0.692 | 0.563 | 11 | 6 | 5 |
| CNRM-CM5 | 0.480 | 0.434 | 0.361 | 0.425 | 31 | 34 | -3 |
| CSIRO-Mk3-6-0 | 0.273 | 0.594 | 0.750 | 0.539 | 15 | 12 | 3 |
| EC-EARTH | 0.823 | 0.427 | 0.506 | 0.585 | 8 | 16 | -8 |
| FGOALS-g2 | 0.607 | 0.600 | 0.569 | 0.592 | 7 | 10 | -3 |
| FIO-ESM | 0.462 | 0.524 | 0.446 | 0.477 | 24 | 25 | -1 |
| GFDL-CM3 | 0.714 | 0.398 | 0.231 | 0.448 | 28 | 28 | 0 |
| GFDL-ESM2G | 0.673 | 0.416 | 0.720 | 0.603 | 4 | 8 | -4 |
| GFDL-ESM2M | 0.643 | 0.514 | 0.681 | 0.613 | 3 | 5 | -2 |
| GISS-E2-H | 0.319 | 0.556 | 0.416 | 0.430 | 29 | 30 | -1 |
| GISS-E2-R | 0.253 | 0.532 | 0.418 | 0.401 | 33 | 32 | 1 |
| HadGEM2-AO | 0.651 | 0.626 | 0.416 | 0.564 | 10 | 9 | 1 |
| HadGEM2-CC | 0.531 | 0.608 | 0.413 | 0.517 | 19 | 17 | 2 |
| HadGEM2-ES | 0.514 | 0.656 | 0.487 | 0.552 | 13 | 11 | 2 |
| inmcm4 | 0.464 | 0.561 | 0.226 | 0.417 | 32 | 27 | 5 |
| IPSL-CM5A-LR | 0.426 | 0.566 | 0.416 | 0.469 | 26 | 20 | 6 |
| IPSL-CM5A-MR | 0.382 | 0.326 | 0.490 | 0.399 | 34 | 35 | -1 |
| IPSL-CM5B-LR | 0.144 | 0.630 | 0.275 | 0.350 | 35 | 31 | 4 |
| MIROC-ESM-CHEM | 0.532 | 0.427 | 0.657 | 0.539 | 2 | 14 | -12 |
| MIROC-ESM | 0.589 | 0.442 | 0.654 | 0.561 | 12 | 15 | -3 |
| MIROC5 | 0.685 | 0.551 | 0.646 | 0.627 | 16 | 2 | 14 |
| MPI-ESM-LR | 0.395 | 0.532 | 0.566 | 0.498 | 21 | 21 | 0 |
| MPI-ESM-MR | 0.437 | 0.513 | 0.569 | 0.506 | 20 | 22 | -2 |
| MRI-CGCM3 | 0.381 | 0.319 | 0.584 | 0.428 | 30 | 33 | -3 |
| NorESM1-M | 0.794 | 0.663 | 0.656 | 0.704 | 1 | 1 | 0 |

It is needed to give some background knowledge about Random Forest method. Why is it selected? It is needed to give some reasoning for this selection. Also in the results and discussion, more explanation is needed for this method.

**Reply**

Thank you very much for this comment. We have now added a new sub-section covering the information related to Random Forest algorithm as show below. .

**3.5.2 Random Forest (RF)**

[revised manuscript text omitted]

(2)

Where $x_{sim,i}$ and $x_{obs,i}$ are the $i^{th}$ data point in the gridded data and observed data series of a climate variable.

**Comment 5**

To have a better understanding about the site, it would be good to add the location of stations on the map.

**Reply**

Thank you very much for your suggestion. We have revised Figure 1 and included the locations of the stations. Also, we have provided the names of stations in Table 2.

[Figure]

**Figure 1.** The location of Pakistan in central-south Asia and the GCM grid points over the country along with locations of precipitation and temperature observation stations. The names of the stations are given in Table 2.

**Table 2.** Validation of accuracy of GPCC precipitation using NRMSE and *md*

| Station No | Station Name | Precipitation ($P$) | | Maximum Temperature ($T_{max}$) | | Minimum Temperature ($T_{min}$) | |
|---|---|---|---|---|---|---|---|
| | | NRMSE | *md* | NRMSE | *md* | NRMSE | *md* |
| 1 | Karachi | 0.530 | 0.840 | 0.270 | 0.880 | 0.180 | 0.919 |
| 2 | Pasni | 0.470 | 0.890 | 0.310 | 0.840 | 0.260 | 0.879 |
| 3 | Nawabshah | 0.740 | 0.740 | 0.300 | 0.850 | 0.170 | 0.919 |
| 4 | Padidan | 0.590 | 0.780 | 0.190 | 0.920 | 0.150 | 0.939 |
| 5 | Jacobabad | 0.520 | 0.840 | 0.100 | 0.960 | 0.090 | 0.959 |
| 6 | Dalbandin | 0.090 | 0.960 | 0.140 | 0.940 | 0.230 | 0.889 |
| 7 | Kalat | 0.970 | 0.870 | 0.240 | 0.900 | 0.470 | 0.779 |
| 8 | Sibbi | 0.590 | 0.880 | 0.390 | 0.810 | 0.260 | 0.889 |
| 9 | Bahawalnagar | 0.530 | 0.810 | 0.310 | 0.899 | 0.270 | 0.881 |
| 10 | Quetta | 0.750 | 0.760 | 0.240 | 0.890 | 0.120 | 0.949 |
| 11 | Multan | 0.730 | 0.740 | 0.120 | 0.950 | 0.120 | 0.949 |
| 12 | Faisalabad | 0.700 | 0.740 | 0.210 | 0.900 | 0.170 | 0.919 |
| 13 | Lahore | 0.710 | 0.700 | 0.140 | 0.940 | 0.110 | 0.959 |
| 14 | Sargodha | 0.790 | 0.680 | 0.160 | 0.930 | 0.170 | 0.919 |
| 15 | Mianwali | 0.720 | 0.750 | 0.240 | 0.890 | 0.120 | 0.949 |
| 16 | Islamabad | 0.450 | 0.840 | 0.160 | 0.930 | 0.190 | 0.909 |
| 17 | Peshawar | 0.690 | 0.720 | 0.190 | 0.920 | 0.110 | 0.949 |

**Comment 6**

It is also recommended to highlight the limitations of the study in the discussion part

**Reply**

Thanks for your suggestion. We have now added the following paragraph to the manuscript to highlight the limitations of this study and recommendation for future work.

"In this study performance of GCMs was assessed based on their ability to simulate past observed $P$, $T_{max}$ and $T_{min}$ and hence the best performing GCMs were identified and used for the development of MMEs. However, it is found that past and future climate may have a weak association hence it is not necessary that if a GCM performs well in the past will give reliable results in future (Knutti et al., 2010). In other words, the best GCMs selected for the MMEs considering their ability to simulate past climate may not be the best in the future under changing climate (Ruane and McDermid, 2017;Ahmed et al., 2019b). This is due to the large uncertainties associated with GHG emission scenarios and GCMs. As a solution to this limitation, Salman et al. (2018) selected an ensemble of GCMs based on past performance as well as the degree of agreement between their future projections. The study detailed in

the present manuscript can be repeated in future to select GCMs considering their past performance and the degree of agreement in their future projections.

In the present study, the MME of $P$, $T_{max}$ and $T_{min}$ were developed by considering top four ranked GCMs. In the past, MMEs were developed considering 3 to 10 top ranked GCMs. However, none of the study showed the performance of MME by varying the number of GCMs in MME. The performance of an MME can be sensitive to the choice of the number of GCMs. Hence, in future, a study should be conducted to investigate the impact of the number of GCMs used for the development of the MME.

Only RF algorithm was used in this study for the development of MMEs. Other machine learning algorithms (e.g. Artificial Neural Networks, Support Vector Machine, Relevance Vector Machine, K-nearest neigbour, Extreme Learning method) can also be used for the development of MMEs. A comparison of the performance of MMEs developed with different machine learning algorithms can assist in identification of pros and cons of different algorithms in relation to development of MMEs.

In the present study, GCM ranking and MME development was conducted only considering $P$, $T_{max}$ and $T_{min}$ pertaining to annual, monsoon, winter, pre-monsoon and post-monsoon seasons. However, several studies reported that the ranking of GCMs based on different climate variables may assist in the identification of a more dependable set of GCMs for ensemble generation (Johnson and Sharma, 2012;Xuan et al., 2017). In future, the ranking of GCMs can be conducted considering several climate variables (e.g. precipitation, mean temperature, maximum temperature, minimum temperature, wind speed, evapotranspiration and solar radiation)."

**Comment 7**

For figures 4, 5, and 7 a performance measure such as r-squared is needed for each scatter plot of observed vs simulated data points.

**Reply**

Thanks for your suggestion. We have indicated the performance of MMEs in terms of the modified index of agreement (*md*) in all plots in Figures 4, 5 and 7 as shown below.

[revised manuscript text omitted]

---

## Author Comment (AC3) · 26 Aug 2019

Comment The authors have evaluated precipitation simulation of 20 CMIP5 GCMs for Pakistan, and developed multi-model ensemble mean at annual and seasonal timescales. The topic is relevant to the journal and the findings are interesting. The authors have shown the application of random forests for MME, which is somehow novel. However, the study needs substantial improvement in explaining the methods. The details of some of the methods are missing and should be further explained. Please find more detailed comments in the following: P refers to page number and L is the line number (please consider using continuous line numbering in future publications)

Reply Thank you very much for your highly constructive comments on our manuscript. We have addressed all your major concerns and minor comments carefully in the revised manuscript. We hope that you will find the revised paper suitable for publication.

Comment 1 P1, L13: "number metrics" Âż "number of metrics"

Reply Thanks. Corrected as suggested.

Comment 2 P1 L14: "very little attention has been given to spatial performance of GCMs" Âż Better to rephrase this sentence, since several studies have considered both spatial and temporal characteristics for evaluating GCMs.

Reply Thanks for your suggestion, we have rephrased it accordingly.

Comment 3 P2, L4: "land and ocean temperature" Âż "land and ocean surface air temperature"

Reply Thanks. Corrected as suggested.

Comment 4 P2, L26: "better GCMs are assigned higher weightages" Âż "higher weights are assigned to better GCMs"

Reply Thanks. Corrected as suggested.

Comment 5 P2, L31: "is climate change modelling" Âż "in climate change modelling"

Reply Thanks. Corrected as suggested.

Comment 6 P3, L7: "such as such as" Âż please remove the redundant "such as".

Reply Thanks. Corrected as suggested.

Comment 7 P5, L25: "five : : : measures" Âż six measures are introduced here. Please revise the number.

Reply Thanks. Corrected as suggested.

Comment 8 P6, L4-5: did you apply these measures on each grid? Or are they applied
on temporally averaged data? For instance, how is KGE calculated? Please explain.

Reply Thanks for your comment. All spatial metrics were applied to each grid point separately and then by spatially averaging them a single value for each spatial metric was obtained. For example, first, KGE was applied to 35 grid points (shown in Figure 1) individually, and then these KGE values obtained for each of these grid points (altogether 35 values) were averaged to obtain a single value representative of the whole study area. We have addresses the issue in section 3.2 of revised manuscript as below: "3.2 GCM Performance Assessment SPAtial EFficiency, Fractions Skill Score, Goodman–Kruskal's lambda, Cramer's V, Mapcurves, and Kling-Gupta efficiency were individually applied to mean annual, monsoon, winter, pre-monsoon, and post-monsoon precipitation, maximum and minimum temperature for each year from 1961 to 2005 of. Later, the GOF values corresponding to each year were temporally averaged to obtain a value for the entire study area."

Comment 9 P5, L8-9: "comprehensive rating metric" Âż what does this indicate? How were the ranks of GCMs (from different measures) combined? Do you mean like averaging the ranks? If so, "comprehensive" is misleading and it is better to be revised.

Reply Thanks for your comment. Comprehensive rating metric is an index used in several studies (Chen et al., 2011; Jiang et al., 2015; Jiang et al., 2012) to combine ranks of a GCM obtained using different metrics or/and considering different climate variables or/and seasons into one single overall rank. In other words, comprehensive rating metric helps to obtain an overall rank from different ranks obtained using different metrics or/and considering different climate variables or/and seasons. In the application of rating metric ranks obtained using different metrics or/and considering different climate variables or/and seasons not averaged to obtain an overall rank for a given GCM. In this study, application of rating metric involved summing the ranks of a GCM corresponding to different seasons (i.e. annual, monsoon and winter) were aggregated and normalized by the refers to the number of GCMs (m = 36) x the number of metrics (n = 6) as shown in Eq 15. The details of the rating matric are given in section

3.3. We agree that word comprehensive rating metric is somewhat misleading, but we have followed the original manuscript where they used the term "Comprehensive Rating Metric".

Comment 10 P6, L28: Eq (1) seems to be the equation for KGE. Please make sure to provide the equation for SPAEF here.

Reply Sorry for this mistake, this should be SPAEF instead of KGE. We have changed it accordingly.

Comment 11 Section 3.1.1: I am still not sure how the measure is calculated? Is it applied to each grid (2°x2°), and then maybe the spatial mean value of KGE is considered? Or, did you take the long-term average of precipitation and then calculate the KGE for a few grids?

Reply Thanks for your comment. In calculating KGE and other metrics, first all monthly data (i.e. GPCC and GCM) of different spatial resolutions (see Table 1 for resolutions) were used to derive data for annual, monsoon, winter, pre-monsoon and post-monsoon seasons corresponding to a common grid with a spatial resolution of 2°x2°. This grid contained 35 points as shown in Figure 1. Then, the means of GPCC and GCM precipitation for the each year for the period 1961 to 2005 were calculated for each grid point for each season and each variable. Later, the GOF values of each year were temporally averaged to obtained a value for the entire study area. We have discusses this in section 3.2 as shown below. "3.2 GCM Performance Assessment SPAtial EFficiency, Fractions Skill Score, Goodman–Kruskal's lambda, Cramer's V, Mapcurves, and Kling-Gupta efficiency were individually applied to mean annual, monsoon, winter, pre-monsoon, and post-monsoon precipitation, maximum and minimum temperature for each year from 1961 to 2005 of. Later, the GOF values corresponding to each year were temporally averaged to obtain a value for the entire study area."

Comment 12 Section 3.1.5 is not clearly explained. A, B, and C need more explanation. What do you mean by "total area of historical and GCM simulated maps"? Is this the

coverage area? If so, then both GCM and obs have the same number of grids and the areas should be identical in all cases. In addition, what does "the degree of intersection (for C)" refer to? How can one quantify such thing? Is there a function for calculating it? These need to be clearly explained.

Reply Thanks for your comment. Total area refers to the historical and GCM simulated maps area. Yes, observed and GCM maps have same number of grid points (i.e. 42) in all cases, C is the intersecting area, and mapcurve function is used in R to calculate GOF values. In order to avoid the confusion, we have revised the explanation as follows. 3.2.5 Mapcurves Mapcurves is another statistical measure, developed by Hargrove et al. (2006) for the measurement of similarity between categorical maps. Mapcurves quantifies the degree of concordance between two maps. The value of Mapcurves can vary from 0 to 1 (perfect agreement). In the present study, the degree of concordance between the historical observed P, Tmax and Tmin map and each of the GCM simulated P, Tmax and Tmin maps was determined using Eq. (11) where, MC_X refers the Mapcurves value, A is the total area of a given category X on the map being compared (i.e. map of a GCM simulated variable), B is the total area of a given category Y on the map of observations, C is the overlapping area between X and Y when the maps are overlaid and n is the number of classes in the reference map (e.g. map of observations).

Eq. (11) (See the supplement file)

In this study the function "mapcurves(x,y)" available in "sabre" package (Nowosad and Stepinski, 2018) written in R programing language was used for estimating mapcurves values. In the above function x, and y are numerical vectors, which represent categorical values of observed precipitation (i.e. GPCC precipitation) or map of observed precipitation and categorical values of GCM simulated precipitation, respectively.

Comment 13 P11, l2: "can be reduce" Âż "can be reduced"

Reply Thanks. Corrected as suggested.

[Figure]

Comment 14 Section 3.4: There is no explanation about the details of the random forest method. How many trees were included? What are the inputs to the model (time series of 4 selected GCMs?)? Is the model applied separately for each grid, or did you employ a consistent model for the entire study domain? How long is the training and testing periods? How did you evaluate the performance of the random forest outputs?

Reply Thanks for the comment; we have added new section 3.5.2 containing information related to Random Forest as shown below.

"3.5.2 Random Forest (RF) Random Forest (RF) algorithm (Breiman, 2001) was used in the calculation of the mean time series of P, Tmax and Tmin corresponding to an MME of four top ranked GCMs. RF is a relatively new machine learning algorithm widely used in modelling non-linear relationships between predictors and predictands (Ahmed et al., 2019b). RF algorithm is found to perform well with spatial data sets and less prone to over-fitting (Folberth et al., 2019). Most importantly Folberth et al. (2019) reported that RF is less sensitive to multivariate correlation. RF is an ensemble technique where regression is done using multiple decision trees. RF algorithm uses the following steps in regression. A bootstrap resampling method is used to select sample sets from training data. Classification And Regression Tree (CART) technique is used to develop unpruned trees using the bootstrap sample. A large number of trees are developed with the samples selected repetitively from training data so that all training data have equal probability of selection. A regression model is fitted for all the trees and the performance of each tree is assessed. Ensemble prediction is estimated by averaging the predictions of all trees which is considered as the final prediction.

Wang et al. (2017a) and He et al. (2016) reported that the performance of RF varies with the number of trees (ntree) and the number of variables randomly sampled at each split in developing the trees (mtry). It was observed that RF performance increases with the increase in ntree. However, in the present study the performance was not found to increase significantly in term of root mean square error when the ntree was greater than 500. Therefore, ntree was set to 500 while the mtry was set to p/3 where p is

the number of variables (i.e. GCMs) used for developing RF-based MME. The MME prediction can be improved by assigning larger weight to the GCMs which show better performance (Sa'adi et al., 2017). RF regression models developed using historical P, Tmax and Tmin simulations of GCMs as independent variable and historical observed P, Tmax and Tmin as dependent variable provide weights to the GCMs according to their ability to simulate historical observed P, Tmax and Tmin. The "Random Forest" package written in R programming language was employed in this study for developing RF-based MMEs. RF-based MMEs were calibrated with the first 70% of the data and validated with the rest of the data.

Comment 15 Figure 4 caption: Does the figure show long-term average values for different grids? Or, does it show spatial mean precipitation in various years. Please clarify it in the caption and the text, and mention the period for it as well.

Reply Thanks for the comment. The Figures 4, 5 and 7 show long-term average values of precipitation for the period 1961 to 2005 over different grids. Each figure has 35 points representing the long-term average values for the whole study area. We have revised the captions as follows.

"Figure 4. Scatter of spatially averaged annual P, Tmax and Tmin, of four top ranked GCMs against GPCC P, CRU Tmax and CRU Tmin for the period 1961 to 2005.

Figure 5. Scatter of spatially averaged annual P, Tmax and Tmin of four lowest ranked GCMs against GPCC P, CRU Tmax and CRU Tmin for the period 1961 to 2005.

Figure 7. Scatter of spatially averaged mean annual GPCC P, CRU Tmax and CRU Tmin MME of four top ranked GCMs against P, CRU Tmax and CRU Tmin using Simple Mean (SM) and Random Forest (RF) for the period 1961 to 2005."

Please also note the supplement to this comment:
https://www.hydrol-earth-syst-sci-discuss.net/hess-2018-585/hess-2018-585-AC3-supplement.pdf

---

## Author Comment (AC4) · 26 Aug 2019

General Comments In this manuscript, authors evaluated precipitation data from 20 CMIP5 GCMs and selected four better-performing CMIP5 GCMs based on their spatial performance against observed precipitation (GPCC) during the historical period (1961-2005). To evaluate the skill of model precipitation (CMIP5 GCMs) against observed precipitation (GPCC), they used six spatial metrics (SPAEF, Goodman-Kruskal's lambda, Fractions Skill Score, Cramer's V, Mapcurves, and Kling-Gupta efficiency). Finally, they generated multi-model ensemble mean (MME) of precipitation of four selected GCMs using Random forest regression and simple mean method. The

manuscript is written fairly well, and the idea of spatial assessment of CMIP5 GCMs for multi-model ensemble mean is appreciated. However, the execution of manuscript seems sloppy and hasty. There are numerous methodological, data, explanation, reporting, and citation issues in the manuscript. Thus I recommend major revisions be required before publication. Reply Thank you for your highly constructive comments and suggestions on our manuscript. Your constructive comments and suggestions helped us to improve the quality of the paper. We have carefully addressed all your comments in the revision of the paper. Revised text is highlighted in red.

Major issues: Comment 1 Error and unexplained parameters in the formula of matrices: I have many doubts about spatial assessment methods. Authors need to explain all six methods clearly and correctly. a) In Goodman-Kruskal's lambda, how many classes you have taken in the contingency matrix? Please mention the number of classes and explain- Are these classes sufficient to explain spatial variability of rainfall or measure the matrix accurately? Did you consider only one annual map to estimate the lambda value for each model? If yes, then there may be many years those have low or high bias but not captured in the annual mean map. You need to estimate lambda value for each year or seasonal map. What is the maxj (or maxj)? What is the value of m and n? Reply a: Thanks for your comment. We have considered seven classes (categories) in the contingency matrix following the study by Demirel et al. (2018). We have addressed the above issues as follows. "Goodman–Kruskal's lambda also known as Lambda coefficient ($\lambda$) is used to measure the nominal/categorical association between categorical maps (Goodman and Kruskal, 1954). Lambda coefficient ($\lambda$) varies between 0 and 1, where a value closer to 1 refers to a higher similarity between the map of model simulations and that of observations of P, Tmax and Tmin. The Lambda ($\lambda$) coefficient was calculated using Eq. (9), where ãĂŰmaxãĂŮ_j is the number of classes (categories) in observed and simulated maps, $c_{ij}$ is a contingency matrix (describes the relationships between the data classes), i and j are the classes in observed and simulated maps, m represent the number of classes in observed and simulated maps respectively. In the present study, seven classes in the contingency matrix were used

by following the study by Demirel et al. (2018). The "DescTools" package (Signorell, 2016) written in R programming language was employed in this study for estimating the nominal/categorical association between observed and simulated maps.

Eq. (9) (see the supplement file)

Regarding the calculation of Lambda value, we have calculated the Lambda value for year and seasons separately and then an average value was considered for the whole study area. We have addressed the above issue in section 3.1 of the revised manuscript as follows. 3.2 GCM Performance Assessment "SPAtial EFficiency, Fractions Skill Score, Goodman–Kruskal's lambda, Cramer's V, Mapcurves, and Kling-Gupta efficiency were individually applied for each year from 1961 to 2005 to mean annual, monsoon, winter, pre-monsoon, and post-monsoon precipitation, maximum and minimum temperature. Later, the GOF values of each year were temporally averaged to obtain a value for the entire study area. The details of the above spatial metrics are given below."

b) In the fraction skill score, there should Nx*Ny in the palace of N. Roberts and Lean,(2008) used Nx*Ny. It will affect the final results. Please explain it. Reply b: Thanks for the comment, we used "verification" package (Pocernich, 2006) written in R programming language for the calculation of FSS. Verification package follows the equations used in Roberts and Lean (2008). Therefore, we have revised FSS equations as below.

3.2.2 Fractions Skill Score The Fractions Skill Score (FSS) proposed by (Roberts and Lean, 2008) is another measure used for the assessment of spatial agreement between model simulations and observations. FSS varies between 0 and 1 where a value closer to 1 refers to higher agreement between observed and simulated data. In this study, FSS between observed and GCM simulated data was computed using Eq. (6).

Eq. (6) (See the supplement file)

[Figure]

In Eq. (6) MSE refers mean square error and is calculated using Eq. (7) and (8).

Eq. (7) (See the supplement file)

Eq. (8) (See the supplement file)

In Eq. (7) and (8) $N_x$ is the number of columns, $N_y$ is the number of rows in a map (observed or simulated), O and M are observed and simulated data fractions respectively. The "verification" package (Pocernich, 2006) written in R programming language was employed in this study for estimating FSS values.

c) In Cramer's V, you have taken the wrong formula. There should be $N*(\min(m-1,n-1))$, but you have taken $N*(\min(m,n)-1$. It will also affect your final selection. Reply c: We agree that Cramér's V is computed by taking the square root of the chi-squared statistic divided by the sample size and the minimum dimension minus 1. Both these expression do the same action. Let's assume m=34 and n=32, thus $\min(m,n)-1=31$ and also $\min(m-1,n-1)=31$. The equation of Cramer's V used in the present study is also same as the one used in the study by Rees (2008). The text was revised as follows.

"Cramer's V (Cramér, 1999) statistic is a Chi-square-test-based measure which is used in assessing spatial agreement between observations and model simulations (Zawadzka et al., 2015). Its value ranges between 0 and 1 and a closer the value to 1 the better the agreement. Cramer's V is calculated using Eq. (10).

Eq. (10) (See the supplement file)

). The "DescTools" package (Signorell, 2016) written in R programming language was employed in this study for calculating Cramer's V values."

d) In Mapcurves method, did you classify your map in the different range of rain? If yes, how many classes you have taken? Did you calculate Y value for each month/ season/year? It should be calculated for each year (1961-2005) between model and GPCC data in the case of annual values. Reply d: Yes, we have classified our data into seven classes and calculated the map curve value for each year and each season and later an average value was considered for the whole study area. We have addressed the issue as follows. "Mapcurves is another statistical measure, developed by Hargrove et al. (2006) for the measurement of similarity between categorical maps. Mapcurves quantifies the degree of concordance between two maps. The value of Mapcurves can vary from 0 to 1 (perfect agreement). In the present study, the degree of concordance between the historical observed P, Tmax and Tmin map and each of the GCM simulated P, Tmax and Tmin maps was determined using Eq. (11) where, ãĂŰMCãĂŮ_X refers the Mapcurves value, A is the total area of a given class X on the map being compared, B is the total area of a given class Y on the observed map, C is the interesting area between X and Y when the maps are overlaid and n is the number of classes in the reference map.

Eq. (11) (See the supplement file)

In this study the function "mapcurves(x,y)" available in "sabre" package (Nowosad and Stepinski, 2018) written in R programing language was used for estimating mapcurves values. In that function x, and y are vectors representing categorical values of categorical values of historical observed data (e.g. GPCC precipitation) and categorical values of simulated data by a GCM, respectively."

e) In Kling-Gupta efficiency, please check Demirel et al., 2018 paper. They have taken different formulas for beta and gamma. Reply e: Thanks, we rechecked the equations with the original paper related to Kling-Gupta Efficiency and found that the equations in our manuscript are correct.

f) Why did you choose these six methods? What are the limitations of each method? Please explain. Reply f: The study by Rees (2008) inspired us to test different spatial metrics in our GCM selection study. Furthermore, these metrics have been also used in other studies (Demirel et al., 2018;Koch et al., 2018;Rees, 2008). We have added a line on page 2, line 15 of introduction section as follows. "These metrics were selected based on their recent applications in spatial performance assessment of models

(Demirel et al., 2018;Koch et al., 2018;Rees, 2008)." The limitations of these metrics are reported in Demirel et al. (2018) as follows. "SPAEF is noted as very discriminative metric in selecting different raster maps whereas other metrics e.g. FSS, Cramer's V (Demirel et al., 2018;Koch et al., 2018) are tolerant (less sensitive). This leads to different results in the spatial calibration of models.

Comment 2 Error in rating metrics formula: (P10, L10) In this formula, rank varies from 1 to 6 (n=6) but it should be 1 to 20 (model=20) for each matrix. Please explain this.

Reply Sorry for the mistake; we have made the necessary correction as follows.

"The overall ranks of GCMs based on different GOFs were obtained for each season separately using Eq. (15).

Eq. (15)

In Eq. (15), n refers to the number of GCMs, m refers to the number of metrics or seasons and i refers to the rank of a GCM based on ith GOF. A value of RM near to 1 refers to a better GCM in terms of its ability to mimic the spatial or temporal characteristics of observations."

Comment 3 Pre-monsoon and Post-monsoon seasons: Why did you not consider the pre and post monsoon season for the analysis and during the overall rank. These seasons will affect significantly in the overall ranking. I recommend to estimate rank month-wise That will improve the results significantly and should not provide the same weight to each month. Here, you provided the same weight to annual, monsoon, and winter rank (during overall rank). Why? Reply Thanks for your suggestion, we have revised whole analysis by considering pre-monsoon and post monsoon along with annual, monsoon and winter seasons. Besides precipitation, we also included maximum and minimum temperature for the selection of GCMs. When different seasons and climate variable were considered it significantly changed the ranks. Large uncertainties are associated in GCM outputs at monthly or finer timescales. Therefore, selections

of GCMs are generally not done based on month-wise ranking. GCMs are generally ranked based on their capability of producing present-day annual and seasonal climatology. This has been mentioned in the revised manuscript as follows.

"GCMs are faltered by the uncertainty in their outputs at monthly or finer timescales such as daily or sub-daily (Xue et al., 2007;Onyutha et al., 2016) (Xue et al., 2007; Onyutha et al., 2016). Therefore, the performances of GCMs are generally evaluated according to their capability of producing present-day mean seasonal cycles, interannual variability, and spatial distribution of climatology at regional or local scales (Meher et al., 2017;Das et al., 2018) (Miao et al. 2012; Fu et al. 2013; Das et al. 2016; Meher and Das, 2017)."

Comment 4 Inconsistency in spatial resolution: You should consider the same spatial resolution to compare the maps or data sample. In the manuscript, observation data (GPCC) are available at 0.5◦ resolution and model data are prepared at 2◦ resolution. Model data should be regridded at 2◦. Reply In order to avoid the confusion, we have added following text to section 2.2.1 of the revised manuscript as follows. "Monthly precipitation data simulated by the 36 CMIP5 GCMs for ensemble member r1i1p1 run were extracted from the IPCC data distribution center (http://www.ipcc-data.org/sim/gcm_monthly/AR5/Reference-Archive.html) for period 1961-2005. The modelling centres, names of GCMs and spatial resolution of each of the selected GCMs are provided in Table 1. In order to have a common spatial resolution, precipitation, maximum and minimum temperature data obtained from different GCMs and GPCC and CRU databases were interpolated into a common 2o×2o grid using bilinear interpolation."

Comment 5 Random Forest Method: Please explain the method and weight value. Reply Thanks for the comment, we have added a new section 3.5.2 for Random Forest description as shown below. 3.5.2 Random Forest (RF) Random Forest (RF) algorithm (Breiman, 2001) was used in the calculation of the mean time series of P, Tmax and Tmin corresponding to an MME of four top ranked GCMs. RF is a relatively new
machine learning algorithm widely used in modelling non-linear relationships between predictors and predictands (Ahmed et al., 2019b). RF algorithm is found to perform well with spatial data sets and less prone to over-fitting (Folberth et al., 2019). Most importantly Folberth et al. (2019) reported that RF is less sensitive to multivariate correlation. RF is an ensemble technique where regression is done using multiple decision trees. RF algorithm uses the following steps in regression. 1. A bootstrap resampling method is used to select sample sets from training data. 2. Classification And Regression Tree (CART) technique is used to develop unpruned trees using the bootstrap sample. 3. A large number of trees are developed with the samples selected repetitively from training data so that all training data have equal probability of selection. 4. A regression model is fitted for all the trees and the performance of each tree is assessed. 5. Ensemble prediction is estimated by averaging the predictions of all trees which is considered as the final prediction. Wang et al. (2017a) and He et al. (2016) reported that the performance of RF varies with the number of trees (ntree) and the number of variables randomly sampled at each split in developing the trees (mtry). It was observed that RF performance increases with the increase in ntree. However, in the present study the performance was not found to increase significantly in term of root mean square error when the ntree was greater than 500. Therefore, ntree was set to 500 while the mtry was set to p/3 where p is the number of variables (i.e. GCMs) used for developing RF-based MME. The MME prediction can be improved by assigning larger weight to the GCMs which show better performance (Sa'adi et al., 2017). RF regression models developed using historical P, Tmax and Tmin simulations of GCMs as independent variable and historical observed P, Tmax and Tmin as dependent variable provide weights to the GCMs according to their ability to simulate historical observed P, Tmax and Tmin. The "Random Forest" package written in R programming language was employed in this study for developing RF-based MMEs. RF-based MMEs were calibrated with the first 70% of the data and validated with the rest of the data.

Comment 6 Increase the number of CMIP5 models in the study: Authors used only 20 models for the current study and said all four RCP data available for 20 models.

**HESSD**

However, there is no use of RCP data in the analysis. Hence, they can get historical data for more than 35 CMIP5 GCMs. That will increase the scope and use of this study. I recommend they should use the maximum number of models. Reply Thanks for your suggestion. We have revised whole analysis by considering precipitation, maximum and minimum temperature data obtained from 36 CMIP5 GCMs.

Comment 7 Selection of better performing models should be based on at least precipitation and temperature: In the manuscript, authors used only precipitation variable to select better performing models, but there are many models under CMIP5 those have low projection skill in temperature data and high skill in precipitation. Hence, there is a possibility of the poor skill of temperature projection in the selected GCMs. Moreover, most of the studies in the hydrology and earth science commonly use precipitation and temperature variables. Therefore, they should include the temperature variable in the analysis and select the models based on the high skill in both (Precipitation and temperature) variables. Reply Following your suggestion, we have selected the GCMs based on annual, monsoon, winter, post and pre-monsoon precipitation, maximum and minimum temperature over Pakistan. The revised results are given below.

[revised manuscript text omitted]

Fig. 7 (See the supplementary file)

Other issues: Comment 1 P2, L1 – please provide citation after several studies (related to the heatwaves, cold snaps etc.). Duffy et al. (2015) is about drought and wet spells. Reply Thanks for your suggestion; we have added citations as given below. "Several studies reported increase in severity and frequency of droughts (Ahmed et al., 2019a), floods (Wu et al., 2014), heatwaves (Perkins-Kirkpatrick and Gibson, 2017)

and decrease in severity and frequency of cold snaps (Wang et al., 2016) in the recent years which are indicative of abrupt variations in the precipitation and temperature regimes."

Comment 2 P2, L7: please provide the correct citation. Hegerl et al.,2018 is not about the affecting hydrological cycle (that include ET, runoff, soil moisture, and precip) Reply Sorry for the mistake, we have changed the reference as given below. The climate modelling community has widely agreed that the sharp temperature rise in the post-industrial revolution era is significantly affecting the global hydrologic cycle (Sohoulande Djebou and Singh, 2015;Evans, 1996).

Comment 3 P2, L9: should be Akhter et al., 2017 Reply Thanks, corrected as suggested.

Comment 4 P2, L10: Wright et al., 2015 is about RCMs. Please provide a correct reference. Reply Sorry for the mistake, we have removed the Wright et al., 2015 citation and added Pour et al., 2018 as below: "Global Circulation Models (GCMs) are principally utilized to simulate and project climate on global scale (Pour et al., 2018;Sachindra et al., 2014).

Comment 5 P2, L13: cite CMIP5 GCMs Reply Thanks, we have added a citation to CMIP5 as shown below. The Coupled Model Intercomparison Project Phase 5 (CMIP5) is a set of GCMs available from the IPCC AR5 (Taylor et al., 2012).

Comment 6 P2, L14: Cited paper is not about the cmip5 and cmip3 comparison. Reply Thanks, reference is replaced as below: "GCMs showed significant improvements in climate simulations compared to its previous generation of CMIP3 models (Gao et al., 2015;Kusunoki and Arakawa, 2015)."

Comment 7 P2, L14: more than 50 GCMS are available. Please check other papers. Reply Thanks for your suggestion we have revised GCM number and citation as below: "Currently, over 50 GCMs are available in the CMIP5 suite with different spatial

resolutions (Hayhoe et al., 2017)."

Comment 8 P2, L16: Ekstrom et al., 2016 is not about size and restriction on the size of the subset of GCMs. Reply We have revised the citation as below: "Human and computational resources pose a restriction on the size of the sub-set of GCMs used in a climate change impact assessment (Herger et al., 2018)."

Comment 9 P2, L16: Salam et al., 2018a and 2018b is same Reply Thanks. Corrected accordingly.

Comment 10 P2, L17: should be 2018 Reply Thanks, corrected as suggested.

Comment 11 P2, L16: cite some paper about the uncertainties in GCMs and why do we need to do ensemble mean. Please add some line about this. Reply Thanks for the comment; we have added some references and text in relation to the above comment. "Sa'adi et al. (2017), Salman et al. (2018), Pour et al. (2018) and Khan et al. (2018) reported that a multi-model ensemble (a sub-set) of GCMs selected considering their skills in reproducing past observed characteristics of climate can reduce the GCM associated uncertainties in climate change impact assessment." Comment 12 P2, L19: "prediction" ("projection") Reply Thanks. Corrected as suggested.

Comment 13 P2, L22: Wang et al., 2017 Reply Thanks. Corrected as suggested.

Comment 14 P2, L24: Wang et al., 2017 Reply Thanks. Corrected as suggested.

Comment 15 P2, L25: Fu et al., 2018 and Dong et al., 2018 are not about the comparison between MME and individual. They are based on temperature projection. Reply Thanks for the comment, we have removed Fu et al., 2018 and Dong et al., 2018 and added two new references as shown below. The SCM is relatively simple to apply and found to perform better than individual GCMs (Weigel et al., 2010;Acharya et al., 2013;Wang et al., 2018).

Comment 16 P2, L31: 2018 Reply Thanks. Corrected as suggested.

Comment 17 P3, L15: Gleckler et al.,2008a and 2018b are same. Reply Thanks. Corrected accordingly.

Comment 18 P4, L1: provide citation after several studies. Reply Thanks for the suggestion; we have provided some references to support our claim. "Overall, review of literature revealed that several studies (Khan et al., 2018;Pour et al., 2018;Salman et al., 2018;Raju et al., 2017) assessed the performance of GCMs considering several grid points over the whole study area; however they ignored the capability of GCMs to replicate the spatial patterns.

Comment 19 P4, L7: you used six methods. Please correct this number throughout the paper. Reply Thanks. Corrected as suggested. Comment 20 P4, L11: please mention the calendar months. Reply Thanks, we have now mentioned the calendar months as shown below. "….assessment of performance of 20 CMIP5 GCM in simulating observed annual (Jan to Dec), monsoon (Jun to Sep) and winter (Dec to Mar), pre-monsoon (Apr to May), and post-monsoon (Oct to Nov) precipitation, maximum and minimum temperature over Pakistan."

Comment 21 Figure 1: should include a climate zone map also. Reply Thanks for the suggestion, we have now provided an aridity map of Pakistan separately as figure 2 adopted from the recent study by (Ahmed et al., 2019d) and added some text in study area section (2.1) as shown below. "Pakistan is overwhelmed by arid and semi-arid climate, and displays significant climatic variations (Figure 2). Figure 2 which is based on the study by Ahmed et al. (2019d) shows that a large area of Pakistan experiences arid climate, followed by semi-arid climate, while a small area in the southwest experiences hyper-arid climate. However, a small area in the top north of the country experiences sub-humid to humid climate.

Fig. 2 (See the supplementary file)

Comment 22 P4, L25-29: this data conflict with the fig 3a. Reply Thanks for the comment. We agree that there is some conflicting information. We checked and found

that Figure 3a is prepared based on 35 grid points at a spatial resolution of 2o x 2o using the GPCC data for the period 1961 to 2005 while the information provided in Line 25 to 29 is based on the study by Ahmed et al. (2017) where they considered 337 grid points over Pakistan for the period 1961 to 2010. Furthermore, Ahmed et al. (2017) classified the precipitation into 10 classes while the present study classified it into seven classes (Figure 3a). We have provided the data period as shown. "The bulk of the summer precipitation is caused by the monsoon winds that arise from the Bay of Bengal while westerly disturbances in the Mediterranean Sea are responsible for the winter precipitation. The average precipitation in Pakistan widely varies from south-west to northern parts in the range of < 100 to > 1000 mm/year during 1961 to 2010. Since the country is mostly characterized by arid and semi-arid climate; the bulk of the country receives less than 500 mm/year of precipitation while only a very limited area in the north receives more than 1,000 mm/year of precipitation (Ahmed et al., 2017)."

Comment 23 P5, L7: please provide the website link (GPCC data). Reply Thanks, we have provided the weblink to GPCC data as shown below. "In this investigation, grid-ded monthly precipitation data of the Global Precipitation Climatology Center (GPCC) (Schneider et al., 2013) (dwd.de/EN/ourservices/gpcc/gpcc.html) were used as the sur-rogates of observed precipitation for the period 1961-2005."

Comment 24 P5, L11: high correlation? Please provide the number. Reply Thanks, we have provided the correlation values as shown below. "Most importantly, GPCC precipitation data have shown correlations above 0.80 with observed precipitation over Pakistan (Ahmed et al., 2019c)."

Comment 25 P5, L14-20: Please mention the ensemble member that you have used in the CMIP5 GCMs. Reply Thanks, we have mentioned the ensemble member as shown below. "Monthly precipitation data simulated by the 20 CMIP5 GCMs for ensemble member r1i1p1 run were extracted. . ..."

Comment 26 P5, L15: provide a website link. Reply Thanks, a web link is provided as shown below. "Monthly precipitation data simulated by the 20 CMIP5 GCMs were extracted from the IPCC data distribution center (http://www.ipcc-data.org/sim/gcm_monthly/AR5/Reference-Archive.html) for period 1961-2005."

Comment 27 P6, L24: Please check the citation. In the introduction, you mentioned Demirel et al., (2018). Reply Sorry for the mistake, we have corrected it as shown below. "SPAtial EFficiency metric (SPAEF), proposed by Demirel et al. (2018) is a robust spatial performance……"

Comment 28 P7, L11: Lambda (heading) Reply Thanks. Corrected as suggested.

Comment 29 P11, L12- 25: You did not mention about the time series. Is it annual rainfall or seasonal or monthly time series? Did you check NRMSE and md between the annual time series? Reply Thanks for your comment. We have revised the section 4.1 as shown below in response to your above comment. "4.1 Accuracy Assessment of Gridded Precipitation Data As a preliminary analysis, the monthly time series of GPCC P, CRU Tmax and CRU Tmin data were validated against the monthly time series of observed P, Tmax and Tmin. The validation was performed for the period 1961-2005. In the present study, two statistical metrics; Normalized Root Mean Square Error (NRMSE), and modified index of agreement (md) were used to assess the accuracy of monthly time series of GPCC P, CRU Tmax and CRU Tmin in replicating the mean and the variability of monthly time series of observed P, Tmax and Tmin. The NRMSE and md values between observed P and GPCC P (pertaining to the grid point closest to the observation station), observed Tmax and Tmin with CRU Tmax and Tmin obtained for 17 locations in Pakistan are given in Table 2. Overall, all the stations showed low and high NRMSE and md values respectively, indicating that the accuracy of the GPCC P in replicating observed precipitation and CRU Tmax and CRU Tmin in replicating observed Tmax and Tmin over Pakistan is high. Overall, NRMSE values were found in the ranges of 0.09 to 0.970 for P, 0.100 to 0.390 for Tmax, and 0.09 to 0.470 for Tmin. At the same time, overall, md values were found in the ranges of 0.680 to 0.960 for P, 0.810 to 0.960 for Tmax, and 0.779 to 0.959 for Tmin.

[Figure]

Table 2 (See the supplementary file)

Comment 30 No need for figure 2. You can remove the figure 2 and include the rank in table 3 in brackets.

Reply Thanks for your suggestion, we have removed Figure 2 and included ranks in Table 3 in brackets as shown below.

Table 3 (See the supplementary file)